# PRIMARY-FINE DECOUPLING FOR ACTION GENERATION IN ROBOTIC IMITATION

**Xiaohan Lei[1], Min Wang[2,*], Wengang Zhou[1,2,†], Xingyu Lu[1], Houqiang Li[1,2]**

[1]MoE Key Laboratory of Brain-inspired Intelligent Perception and Cognition,
University of Science and Technology of China
[2]Institute of Artificial Intelligence, Hefei Comprehensive National Science Center

{leixh, luxingyu010}@mail.ustc.edu.cn
wangmin@iai.ustc.edu.cn, {zhwg, lihq}@ustc.edu.cn

Project Page:   https://xiaohanlei.github.io/projects/PF-DAG

## ABSTRACT

Multi-modal distribution in robotic manipulation action sequences poses critical challenges for imitation learning. To this end, existing approaches often model the action space as either a discrete set of tokens or a continuous, latent-variable distribution. However, both approaches present trade-offs: some methods discretize actions into tokens and therefore lose fine-grained action variations, while others generate continuous actions in a single stage tend to produce unstable mode transitions. To address these limitations, we propose Primary-Fine Decoupling for Action Generation (PF-DAG), a two-stage framework that decouples coarse action consistency from fine-grained variations. First, we compress action chunks into a small set of discrete modes, enabling a lightweight policy to select consistent coarse modes and avoid mode bouncing. Second, a mode conditioned Mean-Flow policy is learned to generate high-fidelity continuous actions. Theoretically, we prove PF-DAG's two-stage design achieves a strictly lower MSE bound than single-stage generative policies. Empirically, PF-DAG outperforms state-of-the-art baselines across 56 tasks from Adroit, DexArt, and MetaWorld benchmarks. It further generalizes to real-world tactile dexterous manipulation tasks. Our work demonstrates that explicit mode-level decoupling enables both robust multi-modal modeling and reactive closed-loop control for robotic manipulation.

## 1 INTRODUCTION

In robotic manipulation, capturing multi-modal distributions in action sequences is essential for learning robust and reliable imitation policies (Florence et al., 2022; Chi et al., 2023). Offline expert trajectories often admit multiple valid actions for the same or similar observations: for example, when an obstacle lies in front of the end-effector, demonstrators may steer either left or right. This richness of valid behaviors complicates learning from offline data and thus motivates the development of various imitation learning approaches to tackle this challenge.

Among these imitation learning approaches, Behavioral Cloning (BC) treats policy learning as supervised regression $a = \pi(o)$ and therefore commonly collapses multiple valid actions into a single mean (Levine et al., 2016; Torabi et al., 2018), as visualized in Figure 1 (a). Action discretization represents multiple modes by predicting categorical bins (Brohan et al., 2022; Zitkovich et al., 2023; Kim et al., 2024), but coarse discretization introduces reconstruction error and temporal discontinuities (Shafiullah et al., 2022), failing to match the smoothness of human demonstrations (see Figure 1 (b)). Generative latent-variable methods instead model $a = \pi(o, z)$ so that sampling different $z$ yields different plausible actions (Zhao et al., 2023; Chi et al., 2023). However, independent per-step resampling of $z$ tends to produce random switches among modes (Chen et al., 2025) (see

---

*Corresponding authors: Min Wang (wangmin@iai.ustc.edu.cn) and Wengang Zhou (zhwg@ustc.edu.cn).

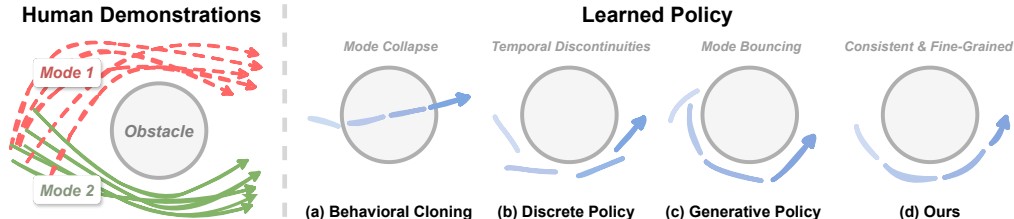

Figure 1: A 2D example illustrating multi-modal expert demonstrations and trajectories predicted by different imitation policies. Behavioral cloning predictions collapse into a single mean. Discrete Policy succeeds but introduces temporal discontinuities. Generative Policy bounces between mode 1 and 2. Our work predicts consistent and fine-grained trajectory.

Figure 1 (c)). Such erratic transitions directly lead to trajectory discontinuities and end-effector pose instability, while further undermining the overall task execution accuracy.

We observe that many manipulation tasks naturally decompose actions into a small set of discrete, interpretable *primary modes* (coarse prototypes such as "lift-and-fold" or "lift-and-rotate") together with continuous, within-mode variations that adjust details like grasp offsets and minor trajectory tweaks. Intuitively, primary modes capture coarse, discrete decisions while within-mode residuals encode fine-grained variations. This observation motivates an explicit separation between i) selecting a coarse, discrete mode consistently and ii) generating the fine-grained continuous action conditioned on that mode.

Motivated by the above, we propose Primary-Fine Decoupling for Action Generation (PF-DAG), a two-stage imitation framework that explicitly separates primary mode selection from continuous action generation. Concretely, PF-DAG first learns a discrete vocabulary of primary modes and a lightweight policy that greedily selects a mode coherently. Then, we introduce a mode conditioned MeanFlow policy, which is a one-step continuous decoder to generate high-fidelity actions conditioned on the selected mode and the current observation. This explicit two-stage decomposition preserves intra-mode variations while reducing mode bouncing by enforcing stable primary choices.

We validate PF-DAG with theoretical and empirical evidence. Among existing methods, single-stage generative policies (Chi et al., 2023; Zhao et al., 2023) are the most direct and competitive end-to-end approach for modeling continuous, multi-modal action distributions, so we focus our theoretical comparison on this family. Under realistic mode-variance assumptions we show that the two-stage design attains a no-higher optimal MSE lower bound than single-stage generative baselines, with a strict improvement whenever the inter-mode variance term is positive. Empirically we test PF-DAG across 56 simulation manipulation tasks (including high-DOF dexterous hands and low-DOF grippers) as well as on real world tactile dexterous manipulation. Results show consistent improvements in accuracy, stability, and sample efficiency compared to diffusion and flow-based baselines, and ablations quantify the contribution of key components. Together, these results suggest that explicitly decoupling coarse discrete decisions from fine-grained continuous generation yields practical and statistical advantages for closed-loop robotic imitation.

## 2 RELATED WORK

### 2.1 BEHAVIOR CLONING

Behavior cloning (BC) casts policy learning as supervised regression on demonstration data (Wang et al., 2017; Torabi et al., 2018; Mandlekar et al., 2021; Hu et al., 2024). In BC, a policy is trained to predict the expert's action for each observed state, yielding a deterministic mapping from states to actions. This approach is highly sample-efficient in practice (*e.g.* for pick-and-place tasks), but it suffers from well-known limitations. In particular, BC policies tend to underfit multi-modal behavior (Mandlekar et al., 2021; Shafiullah et al., 2022; Florence et al., 2022; Chi et al., 2023) and also incur compounding errors at test time (Ross et al., 2011; Ke et al., 2021; Tu et al., 2022; Zhao et al., 2023). To mitigate these issues, recent work has explored more expressive BC models. Implicit BC and energy-based models learn an action-energy landscape per state and solve for actions by optimization (Florence et al., 2022), while mixture-density networks and latent-variable BC attempt to represent multi-modal distributions explicitly (Jang et al., 2022).

## 2.2 DISCRETE POLICY

Discretizing continuous robot actions is viewed as tokenization: converting a high-frequency, high-dimensional control signal into a sequence of discrete symbols so that standard sequence-modeling methods can be applied. Framing actions as tokens has two immediate benefits for manipulation imitation. First, next-token prediction over a discrete vocabulary represents multi-modal conditional action distributions without collapsing modes into a single mean. Second, sequence models bring powerful context modeling and scalable pretraining recipes from language and vision to control, enabling cross-task and cross-embodiment generalization when token vocabularies are shared or aligned. Recent Vision-Language-Action (VLA) efforts articulate this reframing and its practical advantages for large, generalist robot policies (Zitkovich et al., 2023; O'Neill et al., 2024; Kim et al., 2024; Zawalski et al., 2024; Wen et al., 2025; Black et al., 2024; Zheng et al., 2024; Zhen et al., 2024; Cheang et al., 2024; Duan et al., 2024; Zhao et al., 2025).

Existing action tokenizers fall into a few broad families. The simplest and most commonly used approach maps each continuous action dimension at each step to one of a fixed set of bins (Brohan et al., 2022; Zitkovich et al., 2023; Kim et al., 2024). Frequency-space methods like FAST (Pertsch et al., 2025) departs from it and instead compresses action chunks using a time-series transform and lightweight quantization. Others use Vector Quantization (VQ) as latent tokenizers. VQ-based tokenizers learn a shared codebook of action atoms and quantize continuous latent representations to nearest codebook entries (Lee et al., 2024; Wang et al., 2025). While effective at capturing multi-modal action distributions, these approaches inherently trade off reconstruction fidelity for discrete simplicity. Our work differs by leveraging tokenization solely for high-level primary mode selection.

## 2.3 GENERATIVE POLICY

A large class of imitation methods treat policy generation as a stochastic generative problem by introducing latent variables. In this view, a policy is written as $a = \pi(o, z)$ with $z$ sampled from a learned prior. This formulation naturally represents multi-modal conditional action distributions because sampling different $z$ values yields different valid actions for the same observation. Action Chunking with Transformers (ACT) (Zhao et al., 2023) is a sequence generator with Conditional Variational Autoencoder (CVAE) as backend. Diffusion Policy (DP) (Chi et al., 2023) treat action generation as conditional denoising. Starting from noise, the action is iteratively refined via a learned score or denoiser conditioned on observation. More recent normalizing-flow policies (Black et al., 2024; Hu et al., 2024; Zhang et al., 2025) provide tractable density estimation and efficient sampling while representing complex, multi-modal action distributions. Although generative policies represent multi-modal distributions, they often face mode bouncing (Chen et al., 2025), inference cost (Li et al., 2024), chunk trade-offs (Zhao et al., 2023). Other hierarchical approaches, such as Hierarchical Diffusion Policy (HDP) (Ma et al., 2024), also use a high-level policy to guide a low-level generator. However, HDP is designed to rely on explicit, task-specific heuristics like contact-point waypoints to define its hierarchy. In contrast, our PF-DAG learns its primary modes end-to-end directly from action-chunk clusters themselves, offering a more general abstraction not tied to pre-defined heuristics. Thus, we propose to combine the strengths of action tokenization with expressive generative decoders that handle the residual continuous variations. Our PF-DAG decouples the primary discrete mode selection from the fine-grained action generation and reduces mode bouncing while preserving continuous variations.

## 2.4 HIERARCHICAL AND RESIDUAL POLICIES

Our work is also situated within the broader context of hierarchical and residual policies for robot learning (Rana et al., 2023; Cui et al., 2025; Kujanpää et al., 2023; Liang et al., 2024). These approaches commonly decompose the complex control problem into a high-level policy that selects a skill, sub-goal, or context, and a low-level policy that executes control conditioned on the high-level selection (Mete et al., 2024; Feng et al., 2024). For instance, some methods learn residual policies that adapt a base controller (Rana et al., 2023), while others focus on discovering discrete skills from demonstration data or language guidance (Chen et al., 2023; Wan et al., 2024; Tanneberg et al., 2021). While PF-DAG shares this general hierarchical structure, its primary motivation and technical design are distinct. Many hierarchical methods focus on long-horizon planning or unsupervised skill discovery. In contrast, PF-DAG is specifically designed to address the problem of **mode bounc-**

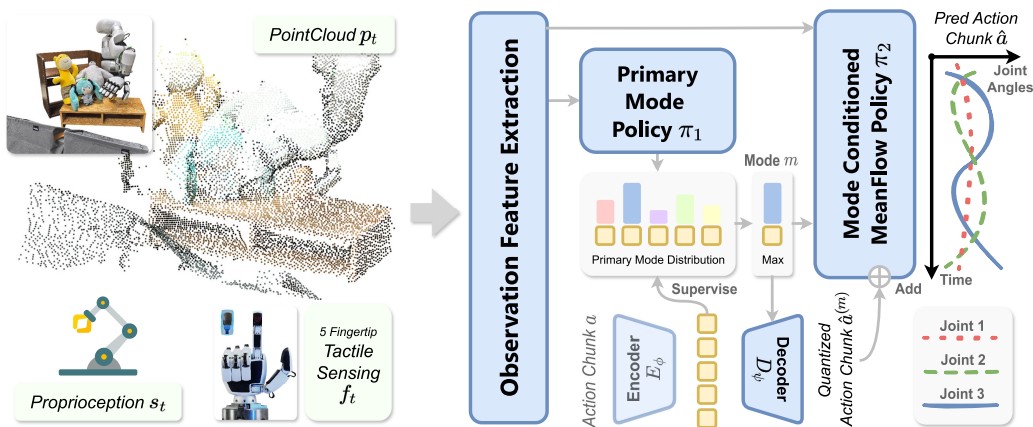

Figure 2: Overview of our PF-DAG framework. The input observation features are extracted via Observation Feature Extraction and then fed to the Primary Mode Policy $\pi_1$. The GT action chunks are compressed into discrete primary modes using VQ-VAE and supervise $\pi_1$, which are only used in training stage. The Mode Conditioned MeanFlow Policy $\pi_2$ takes the selected primary mode $m$ and observation features as input, generating high-fidelity continuous actions.

**ing** inherent in single-stage generative policies when modeling multi-modal action distributions at a fine temporal scale.

## 3  PF-DAG FORMULATION AND DESIGN

This section first defines the task formulation as a closed-loop action-sequence prediction problem, and then presents the three main components of our approach: i) Observation Feature Extraction, ii) a compact discrete representation learned with a Vector-Quantized VAE (VQ-VAE) (Van Den Oord et al., 2017) and a lightweight Primary Mode Policy that predicts those discrete modes, and iii) a mode conditioned one-step continuous decoder based on MeanFlow (Geng et al., 2025). Finally, we give a theoretical analysis that quantifies why a two-stage, coarse-to-fine decomposition reduces the MSE lower bound compared to single-stage generative models.

### 3.1  CLOSED-LOOP ACTION SEQUENCE PREDICTION

Similar to previous work (Chi et al., 2023; Black et al., 2024), we formulate the manipulation task as closed-loop action sequence prediction. Concretely, at time $t$, the observation is $\mathbf{o}_t = (\mathbf{p}_t, \mathbf{s}_t, \mathbf{f}_t)$, where $\mathbf{p}_t$ denotes a fixed-size point cloud, $\mathbf{s}_t \in \mathbb{R}^{d_s}$ denotes robot proprioception, $\mathbf{f}_t \in \mathbb{R}^{5 \times 120 \times 3}$ represents tactile sensing data from the hand's 5 fingertips. For the dimension of $\mathbf{f}_t$, the first dimension 5 corresponds to the 5 individual fingertips, the 120 denotes the number of tactile taxels embedded in each fingertip and the last 3 represents the 3-dimensional force vector. The policy predicts an action chunk $\mathbf{a}_t \in \mathbb{R}^{T_p \times d_a}$ and executes the first $T_a \leq T_p$ steps before re-planning.

$$\hat{\mathbf{a}}_t \sim \pi(\mathbf{o}_t), \qquad \text{execute } \hat{\mathbf{a}}_t[0 : T_a - 1], \text{ then } t \leftarrow t + T_a. \qquad (1)$$

This yields a receding-horizon closed-loop control scheme that preserves temporal coherence and allows fast reaction to new observations. Hyperparameters are presented in Appendix.

### 3.2  OBSERVATION FEATURE EXTRACTION

We first extract the shared observation embedding from input observation $\mathbf{o}_t = (\mathbf{p}_t, \mathbf{s}_t, \mathbf{f}_t)$. Following a simple PointNet-style (Qi et al., 2017) pipeline, each point's coordinates are lifted by an MLP, and LayerNorm (Ba et al., 2016) is applied inside that per-point MLP. Per-point features are aggregated by a symmetric max-pooling. The proprioception $\mathbf{s}_t$ and tactile sensing $\mathbf{f}_t$ are passed through respective MLPs and then concatenated with the point-cloud embedding. A final projection MLP fuses the concatenated vector into the shared observation embedding.

### 3.3 PRIMARY MODE POLICY AND VQ-VAE

Given the shared observation embedding, the framework first selects a primary mode. This subsection describes how we learn a compact VQ-VAE codebook for action chunks and train a lightweight classifier to predict these primary modes from the observation embedding.

**Vector Quantized Variational Autoencoder.** Continuous action chunks $\mathbf{a}$ are compressed into a small discrete set of primary modes $m \in \{1, \ldots, K\}$ using VQ-VAE. Let the deterministic encoder be $E_\phi : \mathbb{R}^{T_p \times d_a} \rightarrow \mathbb{R}^D$ and decoder $D_\psi : \mathbb{R}^D \rightarrow \mathbb{R}^{T_p \times d_a}$. Let the codebook be $\mathbf{C} = \{\mathbf{e}_k \in \mathbb{R}^D\}_{k=1}^K$ with codebook size $K$. We choose $K$ to be small to capture coarse primary action prototypes and make primary policy easy to learn. Given an action chunk $\mathbf{a}$, the encoder produces $\mathbf{z}_e = E_\phi(\mathbf{a})$ and we quantize it to the nearest codebook vector:

$$k^* = \arg \min_k \|\mathbf{z}_e - \mathbf{e}_k\|_2, \qquad \tilde{\mathbf{z}} = \mathbf{e}_{k^*}, \qquad m := k^*. \tag{2}$$

We define $m$ as the primary mode. Reconstruction is $\hat{\mathbf{a}}^{(m)} = D_\psi(\tilde{\mathbf{z}})$. We train the VQ-VAE with the standard commitment and reconstruction terms:

$$\mathcal{L}_{\text{VQ}}(\mathbf{a}) = \|\mathbf{a} - D_\psi(\tilde{\mathbf{z}})\|_2^2 + \|\text{sg}[E_\phi(\mathbf{a})] - \tilde{\mathbf{z}}\|_2^2 + \beta\|E_\phi(\mathbf{a}) - \text{sg}[\tilde{\mathbf{z}}]\|_2^2, \tag{3}$$

where $\text{sg}[\cdot]$ denotes stop-gradient and $\beta$ is the commitment weight. The primary policy is a classifier $\pi_1(m \mid \mathbf{o})$ trained to predict the VQ code $m$ from observation $\mathbf{o}$. At test time, we select the discrete mode $m$ for the current chunk by choosing the highest predicted probability from $\pi_1$. Both encoder $E_\phi$ and decoder $D_\psi$ are implemented as compact MLPs.

**Primary Mode Policy.** The primary policy $\pi_1(m \mid \mathbf{o})$ maps the shared observation embedding to a categorical distribution over the $K$ VQ bins. We implement $\pi_1$ as a lightweight MLP classifier. During training $\pi_1$ is optimized with a standard cross-entropy objective that matches the encoder-assigned VQ indices. At test time we use greedy mode selection for reliability. The separation of primary-mode selection as an explicit classifier drastically reduces coarse mode bouncing.

### 3.4 MODE CONDITIONED MEANFLOW POLICY

After selecting a primary mode $m$, we recover a high-quality continuous action chunk that respects the selected mode. To balance generation quality and real-time responsiveness, we use a one-step generative modeling inspired by MeanFlow (Geng et al., 2025). Instead of multi-step denoising iterations, a learned average velocity field predicts the displacement from noise to the desired action in one function evaluation. Let $m$ be the selected discrete mode and $\hat{\mathbf{a}}^{(m)} := D_\psi(\mathbf{e}_m)$ be the VQ-decoder reconstruction of the mode. The role of the one-step generator is to produce a residual $\Delta\mathbf{a}$ conditioned on observation $\mathbf{o}$ and mode $m$, such that the final action chunk is $\hat{\mathbf{a}} = \hat{\mathbf{a}}^{(m)} + \Delta\mathbf{a}$.

**Mode and Observation Conditioned Average Velocity Field.** Following MeanFlow (Geng et al., 2025), we implement the residual as an average velocity field $\bar{\mathbf{v}}_\theta(\mathbf{z}_r, \tau, r; \mathbf{o}, m)$, where $\mathbf{z}_r$ denotes a state on the interpolation path between noise sample and the target action, $\tau \in [0, 1]$ is the interpolation start time, and $r \in (0, 1]$ is the end time. The MeanFlow field is trained to match the ground-truth average velocity over arbitrary intervals $[\tau, r]$, which is written as

$$\bar{\mathbf{v}}^*(\mathbf{z}_r, \tau, r) = \text{sg}\left(\frac{d\mathbf{z}_r}{dr} - (r - \tau)\left(\frac{d\mathbf{z}_r}{dr}\frac{\partial\bar{\mathbf{v}}_\theta}{\partial\mathbf{z}} + \frac{\partial\bar{\mathbf{v}}_\theta}{\partial r}\right)\right). \tag{4}$$

The $\frac{d\mathbf{z}_r}{dr}$ is the instantaneous velocity of $\mathbf{z}_r$ at time $r$. $\frac{\partial\bar{\mathbf{v}}_\theta}{\partial\mathbf{z}}$ describes how the average velocity responds to perturbations in the residual draft, and $\frac{\partial\bar{\mathbf{v}}_\theta}{\partial r}$ captures how it evolves as the interpolation approaches the target residual. We train $\bar{\mathbf{v}}_\theta$ with squared-error objective that supervises the predicted average velocity. More detailed derivations of the formulation are provided in Appendix.

**Implementation Details.** For backbone modeling we use a DiT-style transformer backbone (Peebles & Xie, 2023). Each action chunk is represented as a sequence of tokens. The time-related scalars $\tau$ and $r$ are expanded via sinusoidal embeddings (Vaswani et al., 2017), which are added to observation embedding, as well as a learnable embedding of the discrete mode $m$. During training, $(\tau, r)$ is sampled from a uniform distribution and $\mathbf{z}_0$ is from standard normal distribution.

### 3.5 THEORETICAL ANALYSIS

With the two-stage architecture defined, we now provide a concise theoretical analysis that explains why this coarse-to-fine decomposition strictly reduces the minimum achievable MSE compared to

| Method | Adroit | | | DexArt | | | | MetaWorld | | Success |
|---|---|---|---|---|---|---|---|---|---|---|
| | Hammer | Door | Pen | Laptop | Faucet | Toilet | Bucket | Medium (6) | Hard (5) | |
| IBC | $0.00_{\pm 0.00}$ | $0.00_{\pm 0.00}$ | $0.10_{\pm 0.01}$ | $0.01_{\pm 0.01}$ | $0.07_{\pm 0.02}$ | $0.15_{\pm 0.01}$ | $0.00_{\pm 0.00}$ | $0.11_{\pm 0.02}$ | $0.09_{\pm 0.03}$ | 0.08 |
| BC-H | $0.10_{\pm 0.09}$ | $0.07_{\pm 0.05}$ | $0.16_{\pm 0.03}$ | $0.09_{\pm 0.02}$ | $0.13_{\pm 0.04}$ | $0.21_{\pm 0.02}$ | $0.10_{\pm 0.01}$ | $0.15_{\pm 0.03}$ | $0.18_{\pm 0.05}$ | 0.15 |
| DP | $0.48_{\pm 0.17}$ | $0.50_{\pm 0.05}$ | $0.25_{\pm 0.04}$ | $0.69_{\pm 0.04}$ | $0.23_{\pm 0.08}$ | $0.58_{\pm 0.02}$ | $0.46_{\pm 0.01}$ | $0.20_{\pm 0.05}$ | $0.19_{\pm 0.03}$ | 0.30 |
| DP3 | $1.00_{\pm 0.00}$ | $0.62_{\pm 0.04}$ | $0.43_{\pm 0.06}$ | $0.83_{\pm 0.01}$ | $0.63_{\pm 0.02}$ | $\mathbf{0.82_{\pm 0.04}}$ | $0.46_{\pm 0.02}$ | $0.45_{\pm 0.05}$ | $0.35_{\pm 0.02}$ | 0.51 |
| FlowPolicy | $1.00_{\pm 0.00}$ | $0.58_{\pm 0.05}$ | $0.53_{\pm 0.12}$ | $0.85_{\pm 0.02}$ | $0.42_{\pm 0.10}$ | $0.80_{\pm 0.05}$ | $0.39_{\pm 0.06}$ | $0.47_{\pm 0.07}$ | $0.37_{\pm 0.07}$ | 0.51 |
| **PF-DAG (Ours)** | $\mathbf{1.00_{\pm 0.00}}$ | $\mathbf{0.65_{\pm 0.03}}$ | $\mathbf{0.65_{\pm 0.01}}$ | $\mathbf{0.90_{\pm 0.02}}$ | $\mathbf{0.72_{\pm 0.05}}$ | $\mathbf{0.82_{\pm 0.02}}$ | $\mathbf{0.47_{\pm 0.02}}$ | $\mathbf{0.68_{\pm 0.04}}$ | $\mathbf{0.72_{\pm 0.03}}$ | **0.72** |

Table 1: Quantitative comparison of PF-DAG against state-of-the-art baselines on 18 tasks from three simulation benchmarks.

single-stage generative predictors. Single-stage generative methods produce actions by sampling a latent code $\mathbf{z} \sim \mathcal{N}(0, I)$ and decoding $\hat{\mathbf{a}}_g = \pi(\mathbf{o}, \mathbf{z})$. Under the squared-error criterion, the best point estimate is the conditional expectation $\hat{\mathbf{a}}_g^*(\mathbf{o}) = \mathbb{E}_{\mathbf{z}}[\pi(\mathbf{o}, \mathbf{z})]$. The resulting expected MSE decomposes into an irreducible data variance term and a model bias:

$$\mathbb{E}_{\mathbf{o},\mathbf{a}}\big[\|\mathbf{a} - \hat{\mathbf{a}}_g^*(\mathbf{o})\|^2\big] = \mathbb{E}_{\mathbf{o}}\big[\mathrm{Var}(\mathbf{a} \mid \mathbf{o})\big] + \mathbb{E}_{\mathbf{o}}\big[\|\mathbb{E}[\mathbf{a} \mid \mathbf{o}] - \hat{\mathbf{a}}_g^*(\mathbf{o})\|^2\big]. \quad (5)$$

When the model is unbiased the second term vanishes and the minimum achievable error equals $\mathbb{E}_{\mathbf{o}}[\mathrm{Var}(\mathbf{a} \mid \mathbf{o})]$.

In our two-stage scheme the primary stage selects a discrete mode $\hat{m}(\mathbf{o})$ and the second stage outputs $\hat{\mathbf{a}}(\mathbf{o}, m, \mathbf{z}) = \pi_2(\mathbf{o}, m, \mathbf{z})$. For any fixed $(\mathbf{o}, m)$, the optimal MSE predictor collapses the stochasticity in $\mathbf{z}$ to the conditional expectation $\hat{\mathbf{a}}^*(\mathbf{o}, m) = \mathbb{E}_{\mathbf{z}}[\pi_2(\mathbf{o}, m, \mathbf{z})]$, yielding the irreducible residual $\mathbb{E}_{\mathbf{o},m}[\mathrm{Var}(\mathbf{a} \mid \mathbf{o}, m)]$ when the model is unbiased. By the law of total variance,

$$\mathbb{E}_{\mathbf{o},m}\big[\mathrm{Var}(\mathbf{a} \mid \mathbf{o}, m)\big] = \mathbb{E}_{\mathbf{o}}\big[\mathrm{Var}(\mathbf{a} \mid \mathbf{o})\big] - \mathbb{E}_{\mathbf{o}}\big[\mathrm{Var}_{m|\mathbf{o}}\big(\mathbb{E}[\mathbf{a} \mid \mathbf{o}, m]\big)\big], \quad (6)$$

which is no greater than $\mathbb{E}_{\mathbf{o}}[\mathrm{Var}(\mathbf{a} \mid \mathbf{o})]$, and is strictly smaller whenever $\mathrm{Var}_{m|\mathbf{o}}\big(\mathbb{E}[\mathbf{a} \mid \mathbf{o}, m]\big) > 0$. Intuitively, discretizing into primary modes removes inter-mode variance from the residual error, lowering the MSE bound compared to single-stage latent samplers.

## 4 EXPERIMENTS

### 4.1 SIMULATION EVALUATION

**Benchmarks and Datasets.** We evaluate our method on manipulation benchmarks that cover a broad range of control domains. We use Adroit (Rajeswaran et al., 2017), DexArt (Bao et al., 2023) and MetaWorld (Yu et al., 2020) as our simulation benchmarks. These are implemented on physics engines like MuJoCo (Todorov et al., 2012) and IsaacGym (Makoviychuk et al., 2021). For fair comparison we adopt the same task splits and data collection pipelines as in prior work (Ze et al., 2024): Adroit tasks with high-dimensional Shadow hand and MetaWorld with low-dimensional gripper are trained with 10 expert demos per task, while DexArt with Allegro hand uses 90 expert demos. Demonstrations are collected using scripted policies for MetaWorld tasks, and RL-trained expert agents (Wang et al., 2022; Schulman et al., 2017) for Adroit and DexArt. Each experiment is run with three random seeds. For each seed we evaluate the policy for 20 episodes every 200 training epochs and then compute the average of the top-5 highest success rates (Ze et al., 2024). The final metric is the mean and standard deviation across the three seeds.

**Experiment Setup.** All networks are optimized with AdamW (Loshchilov & Hutter, 2017). We apply a short linear warmup followed by cosine decay for the learning rate. Training proceeds in stages: first we pretrain the VQ-VAE to learn compact primary prototypes; then we freeze the codebook and jointly train the Primary Mode Policy $\pi_1$ (cross-entropy to the VQ indices) and the mode-conditioned MeanFlow generator $\bar{v}_\theta$ (squared-error supervision on sampled $(\tau, r)$ intervals). At inference we set $(\tau, r) = (0, 1)$ for one-step continuous action chunk generation.

**Baselines.** We compare against the following representative baselines. Implicit Behavioral Cloning (IBC) (Florence et al., 2022) serves as a representative implicit BC method. BC-H (Foster et al., 2024) represents non-generative approaches for mitigating mode instability. Diffusion Policy (DP) (Chi et al., 2023) pioneers the original formulation of image-conditioned diffusion-based policies. While 3D Diffusion Policy (DP3) (Ze et al., 2024) represents a recent advancement in 3D-point-cloud conditioned diffusion-based policies, Flow Policy (FP) (Zhang et al., 2025) falls into the category of normalizing-flow-based policy variants. These baselines provide a spectrum from energy-based model to expressive generative policies.

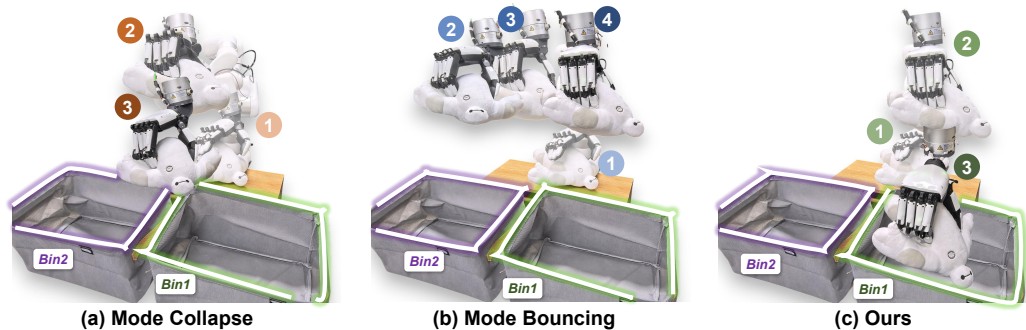

| (a) Mode Collapse | (b) Mode Bouncing | (c) Ours |

Figure 3: Visual comparison of failure modes in baselines versus PF-DAG. **Mode Collapse** outputs "average" actions, while **Mode Bouncing** randomly switches between consecutive time steps.

| Task Description | Task Parameters | | | | | Success Rate | | |
| --- | --- | --- | --- | --- | --- | --- | --- | --- |
| | # Variations | # Demos | End Effector | Tactile | Action Dim | Vanilla BC | DP3 | **PF-DAG (Ours)** |
| *Pick Cube* | 3 | 50 | Gripper | ✗ | $7+1$ | 0.20 | 0.60 | **0.70** |
| *Place Baymax* | 1 | 10 | Gripper | ✗ | $7+1$ | 0.40 | 0.85 | **0.90** |
| *Wipe Table* | 5 | 50 | XHand | ✓ | $7+12$ | 0.00 | 0.55 | **0.70** |
| *Place Toy Into Bin* | 4 | 50 | XHand | ✓ | $7+12$ | 0.00 | 0.60 | **0.80** |

Table 2: Quantitative comparison of success rates of different methods on real world manipulation tasks. The table presents key task parameters alongside the performance of each method.

**Key Findings.** Across Adroit, DexArt and MetaWorld, our method substantially outperforms diffusion and other baselines. Table 1 highlights our work performance on 18 core tasks, while comprehensive results across all 56 tasks are detailed in Appendix. Beyond that, our two-stage design preserves primary-mode consistency even when action chunks are short, which approaches closed-loop and highly reactive operation. Meanwhile, our primary-mode tiny MLP and the one-step generator together yield fast generation while maintaining high success rates, as discussed in Ablations section. These findings indicate that explicitly decoupling coarse discrete mode selection from continuous intra-mode variation yields both statistical and practical benefits.

## 4.2 REAL WORLD EVALUATION

**Hardware.** We evaluate our method on two single-arm hardware configurations commonly used in manipulation research: a) an UFACTORY xArm manipulator [1] equipped with a two-finger parallel gripper, and b) an xArm paired with ROBOTERA XHand [2] for dexterous manipulation. For visual sensing we use a third-person Intel RealSense L515 LiDAR camera that provides aligned color and depth frames. For the *xArm + gripper* setup we additionally use a low-cost 3D-printed demonstration arm from GELLO (Wu et al., 2024) as teleoperation device. For the *xArm + XHand* setup,

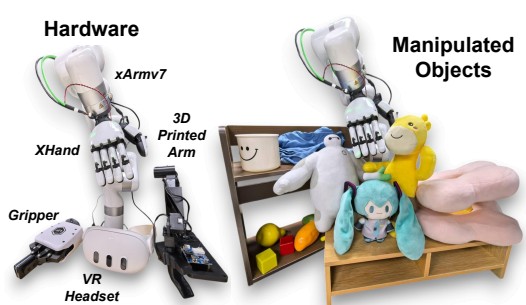

Figure 4: Hardware and manipulated objects used in real world experiments.

human hand motion is captured from a Meta Quest 3 headset and retargeted to the XHand. All computation runs on a single workstation equipped with an NVIDIA RTX 4090 laptop GPU. The robot and sensors are controlled over a local area network.

**Teleoperation.** We collect demonstrations with two teleoperation pipelines. *xArm + gripper* demonstrations are collected using the GELLO framework (Wu et al., 2024), a low-cost and intuitive teleoperation system that allows operators to demonstrate end-effector motions with a separate low-cost manipulator. *xArm + XHand* demonstrations are recorded by capturing human hand kinematics via a Meta Quest 3 headset. The recorded wrist 6-DoF pose is mapped to the xArm end-effector via Inverse Kinematics (IK), finger joint values are retargeted to the XHand via AnyTeleop (Qin et al., 2023).

---

[1] https://www.ufactory.cc  [2] https://www.robotera.com

| Ablations | | | | Benchmarks | | | |
|---|---|---|---|---|---|---|---|
| w. PM | w. MF | Token. | # Modes | Adroit (3) | DexArt (4) | MetaWorld (11) | Weighted Success |
| ✓ | ✓ | VQ-VAE | 64 | **0.77**$_{\pm 0.03}$ | **0.72**$_{\pm 0.04}$ | **0.70**$_{\pm 0.02}$ | **0.72** |
| ✓ | ✗ | VQ-VAE | 64 | 0.02$_{\pm 0.01}$ | 0.00$_{\pm 0.00}$ | 0.02$_{\pm 0.01}$ | 0.01 |
| ✗ | ✓ | VQ-VAE | 64 | 0.62$_{\pm 0.02}$ | 0.65$_{\pm 0.01}$ | 0.51$_{\pm 0.03}$ | 0.56 |
| ✓ | ✓ | VQ-VAE | 8 | 0.72$_{\pm 0.03}$ | 0.70$_{\pm 0.02}$ | 0.55$_{\pm 0.05}$ | 0.61 |
| ✓ | ✓ | VQ-VAE | 1024 | 0.66$_{\pm 0.02}$ | 0.68$_{\pm 0.03}$ | 0.52$_{\pm 0.06}$ | 0.58 |
| ✓ | ✓ | K-means | 64 | 0.76$_{\pm 0.05}$ | 0.70$_{\pm 0.01}$ | 0.69$_{\pm 0.02}$ | 0.70 |

Table 3: Ablation study on the impact of PF-DAG's key components and hyperparameters. **w. PM** denotes whether the primary mode policy is included. **w. MF** indicates whether the mode-conditioned MeanFlow policy is included. **Token.** means action tokenization method. **# Modes** represents the number of discrete primary modes.

**Observation and Action Spaces.** Visual input is the RGB-D stream from the RealSense L515. Frames are backprojected to form a colored point cloud. We convert each frame into a fixed-size point cloud by applying Farthest Point Sampling. Proprioceptive observations include the xArm joint angles. When the XHand is integrated, the observation space is extended to include the XHand's joint angles as additional dimensions. For the XHand configuration we additionally log tactile readings from fingertip sensors. All observations are normalized using the statistics computed on the training split. The policy outputs actions directly in joint space for both setups. We operate in absolute joint position control.

**Baselines.** We compare our method to two baselines. **Vanilla BC** processes observations through the same Observation Feature Extraction pipeline used by our method, and a 3-layer MLP is trained to regress actions in a standard behavior cloning setup (Levine et al., 2016). **DP3** (Ze et al., 2024) is a diffusion-based generative policy operating on 3D point-cloud-conditioned actions. At inference DP3 employs DDIM (Song et al., 2020) denoising to obtain actions. Both baselines are trained on the identical demonstration sets and evaluated under the same closed-loop control as our method.

**Tasks.** We evaluate on tasks spanning low-DOF gripper control and high-DOF tactile dexterous manipulation. Low-DOF examples include a *Pick Cube* task (gripper picks a cube from randomized table locations) and a *Place Baymax* task (place a toy "Baymax" from table into a cabinet). High-DOF experiments use a 12-DOF dexterous hand equipped with tactile sensing and include contact-rich tasks such as *Wipe Table* (multiple possible wiping contact points) and *Place Toy Into Bin* (multiple candidate toy-boxes yielding multi-modal valid outcomes). For task like *Place Baymax* we exploit a pretraining to fine-tuning regime. Models pretrained on a source task require substantially fewer target-task demonstrations to reach competitive performance.

**Result Analysis.** Our method consistently outperforms both baselines in success rate across low-DOF and high-DOF/tactile tasks (see Table 2). Qualitatively, Figure 3 visualizes common failure modes of baselines, while our policy commits to coherent, single-mode rollouts when appropriate and preserves intra-mode variations elsewhere. Typical failure cases for our method occur at out-of-distribution object placements or when tactile sensing is intermittently noisy. These failures are rare and amenable to mitigation via modest additional demonstrations or data augmentation.

### 4.3 ABLATIONS

**Primary Mode and MeanFlow Ablation.** We ablate the two core components of our pipeline to establish their individual importance. First is to remove the Mode-conditioned MeanFlow Policy (MF) so that the system simply uses the Primary Mode Policy (PM)'s predicted VQ code and decodes it via the VQ-VAE reconstruction as the final action. Second is to remove the PM so that MF attempts to predict actions without being conditioned on a discrete mode. Results are reported in Table 3. Removing MF collapses performance almost completely, showing that a raw VQ reconstruction is insufficient as the final action when the number of modes is limited. The quantization error produces large reconstruction distortions that destroy task success. Conversely, removing PM yields a $0.16$ absolute drop in success, which demonstrates that an explicit primary-mode selection substantially eases the downstream continuous generation problem and prevents coarse-mode bouncing.

**Mode Capacity and Tokenization.** We study how the number of discrete primary modes $K$ and the choice of tokenization affect the PM learning and final task performance. We vary $K \in \{8, 64, 1024\}$ and compare VQ-VAE with a k-means tokenization baseline. Results appear

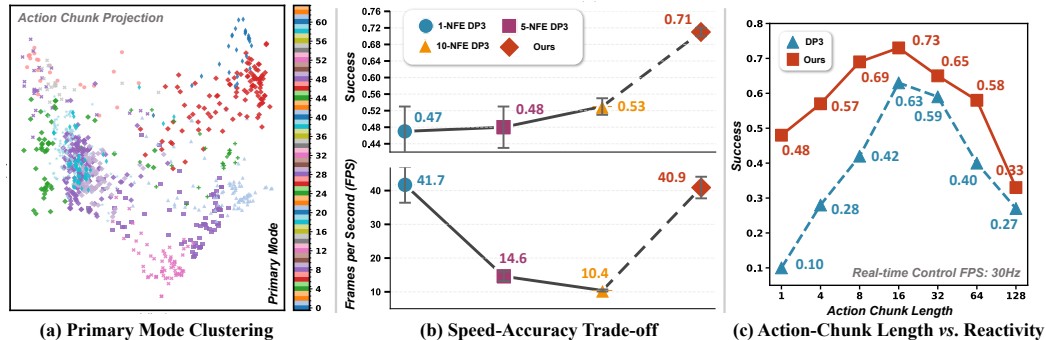

**(a) Primary Mode Clustering**    **(b) Speed-Accuracy Trade-off**    **(c) Action-Chunk Length vs. Reactivity**

Figure 5: Illustration of critical properties of PF-DAG. (a) Action chunks are projected to 2D via PCA, colored by their assigned primary mode. (b) PF-DAG's one-step MeanFlow decoder achieves FPS comparable to 1-NFE DP3 while maintaining significantly higher success. (c) PF-DAG preserves high success even with short chunks by avoiding primary mode bouncing.

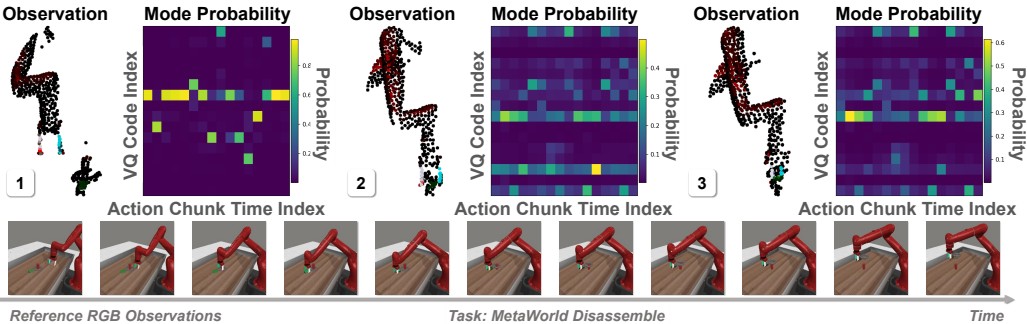

Figure 6: Visualization of the mode probability distribution predicted by the Primary Mode Policy $\pi_1$ at three selected frames. The vertical axis of the heatmap represents the mode index.

in Table 3. A small $K$ helps the PM to learn, but it risks underfitting when the task's action-chunk distribution is complex. Large $K$ increases expressivity but makes the PM hard to learn. In our domains the trade-off is modest and $K = 64$ achieves a good balance across tasks. Different tokenizers produce similar final success rates, suggesting the approach is robust to discretization methods. Our method mainly needs a reasonable set of coarse modes rather than a specific quantizer. To further illustrate mode structure we project action-chunks into 2D (PCA) and color by assigned mode. The visualization shows clear coarse-mode clusters on the action manifold, as visualized in Figure 5 (a). We also visualize the policy's outputs at different timesteps (see Figure 6). For a more detailed qualitative analysis of the primary mode policy's behavior, see Appendix.

**Ablation on Meanflow.** This ablation aims to evaluate the performance difference between MeanFlow and Conditional Flow Matching (CFM) (Lipman et al., 2022), where CFM is tested under different Ordinary Differential Equation (ODE) numerical integration methods. Though CFM theoretically defines a constant velocity field when mapping from noise to target distribution, the parameterized neural network introduces nonlinearity in numerical computations. This makes the exploration of diverse ODE integrators non-trivial. For Runge-Kutta integration, we adopt the Dormand-Prince 5 method, a widely used choice for adaptive-stepsize ODE solving. As shown in Table 4, varying ODE numerical integrators yields negligible performance improvements for CFM. In contrast, replacing CFM with MeanFlow results in a performance gain.

| Variations | ODE Solver | Success |
|---|---|---|
| CFM | 1-NFE Euler | $0.69_{\pm 0.03}$ |
| CFM | 10-NFE Euler | $0.69_{\pm 0.02}$ |
| CFM | Runge-Kutta | $0.68_{\pm 0.01}$ |
| Meanflow | 1-NFE | $\mathbf{0.72_{\pm 0.02}}$ |

Table 4: Comparison of success for Mean-Flow (MF) and Conditional Flow Matching (CFM), varying the ODE solver and NFE.

**Speed–Accuracy Trade-off.** We examine how the number of function evaluations (NFE) during inference affects both inference speed and success. We compare our one-step MeanFlow decoder to DP3 at different NFE settings, plots are in Figure 5 (b). Our one-step generator achieves inference speed comparable to DP3 with 1-NFE while delivering substantially higher success. More

generally, we observe that within the tested range the total NFE has a surprisingly small influence on success, which suggests that for these simulated tasks the NFE is not the dominant bottleneck. We hypothesize this limited sensitivity is due to the tasks' tolerance to small action perturbations in simulation.

**Action-chunk Length and Reactivity.** All experiments here are conducted on real settings described in the Real World Evaluation section. We sweep action-chunk length and measure success. Shorter chunks make the controller more open-loop reactive and therefore better able to respond to unexpected environment changes. However, short chunks also tend to increase trajectory jitter and occasional stoppages. Our method maintains relatively high success even at short chunk lengths, showing the two-stage design preserves primary-mode consistency while allowing rapid reactivity. Results appear in Figure 5 (c).

### 4.4 FAILURE CASES AND LIMITATIONS

While PF-DAG demonstrates strong performance across a wide range of manipulation benchmarks, it has two notable limitations. First, because primary-mode selection operates on discretized action chunks, the method exhibits reduced temporal granularity in very high-dynamics, low-latency tasks. Second, the discrete codebook introduces a trade-off between expressivity and learnability. Larger $K$ improves representational capacity but makes primary-policy learning harder, while smaller $K$ constrains diversity. We address this in the paper via targeted ablations and validation sweeps. Promising directions to reduce per-task tuning include shared or meta-learned codebooks, end-to-end distillation, and multi-task pretraining to improve generalization and reduce pipeline overhead.

## 5 CONCLUSION

In this work we present PF-DAG, a two-stage imitation learning framework that decouples primary mode selection from fine-grained action generation. PF-DAG first uses a VQ-VAE to tokenize action chunks into discrete modes. A lightweight primary policy is then trained to predict these modes from observations, allowing for stable and consistent coarse mode selection. The framework then employs a mode conditioned MeanFlow policy to produce high-fidelity continuous actions conditioned on the selected mode. We prove that, under realistic variance assumptions, PF-DAG attains a strictly lower MSE bound than comparable single-stage generative policies. Empirically, PF-DAG outperforms state-of-the-art baselines on 56 simulated tasks and on real world tactile dexterous manipulation. Future work will extend PF-DAG to long-horizon hierarchical control and investigate uncertainty-aware refiners for improved robustness.

## 6 ACKNOWLEDGMENTS

This work was supported by the National Natural Science Foundation of China under Contract 62472141, the Natural Science Foundation of Anhui Province under Contract 2508085Y040, and the Youth Innovation Promotion Association CAS. It was also supported by the GPU cluster built by MCC Lab of Information Science and Technology Institution, USTC and the Supercomputing Center of USTC.

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
