# OpenReview forum: "Primary-Fine Decoupling for Action Generation in Robotic Imitation"
_ICLR.cc/2026/Conference — ICLR 2026 Poster_

### Official Review · Reviewer_u2ZP · 2025-10-16

**Soundness:** 3
**Presentation:** 3
**Contribution:** 3
**Rating:** 8
**Confidence:** 3

**Summary:**

This paper introduces PF-DAG (Primary-Fine Decoupling for Action Generation), a two-stage imitation learning framework that separates coarse discrete mode selection from fine-grained continuous action generation in robotic manipulation. The method first compresses action chunks into discrete primary modes using a VQ-VAE and predicts these modes through a lightweight classifier π₁ (Sec. 3.3). Conditioned on the selected mode, a one-step MeanFlow policy π₂ produces continuous residual actions (Sec. 3.4). The authors show theoretically (Eq. 6) that this two-stage design achieves a lower optimal MSE bound than single-stage generative models by removing inter-mode variance. Experiments on 56 manipulation tasks across Adroit, DexArt, and MetaWorld (Table 1, Table 5) demonstrate significant improvements over diffusion and flow-based baselines such as DP3 and FlowPolicy. Real-world evaluations on xArm and XHand setups (Table 2) further support PF-DAG’s efficiency and robustness, and ablations (Table 3, Table 7) examine the contributions of each component, the number of discrete modes, and tokenization schemes.

**Strengths:**

1. The paper presents a clean, interpretable two-stage structure that maps well onto the inherent hierarchy of manipulation behaviors. The distinction between discrete primary modes (coarse subgoals) and continuous fine variations (residual control) is intuitive and empirically justified by improved temporal stability and reduced “mode bouncing” (Fig. 3, Fig. 5c).

2. PF-DAG is evaluated on 56 simulated and several real-world tasks, with consistent and sizable gains over strong diffusion and flow baselines (Table 1, Table 5). The inclusion of tactile sensing and dexterous-hand setups makes the evaluation unusually comprehensive for imitation learning studies.

3. The ablation studies (Table 3) systematically isolate the contributions of the Primary Mode Policy and MeanFlow decoder, demonstrating clear complementary effects. Additional analyses on the number of discrete modes (Table 7) and solver sensitivity (Table 4) reveal robustness to hyperparameters and architecture choices.

4. The MeanFlow one-step generation (Sec. 3.4) avoids the high computational overhead of multi-step diffusion or normalizing flow inference, achieving inference speeds comparable to 1-NFE diffusion with substantially higher success (Fig. 5b). This efficiency supports PF-DAG’s practicality for real-time control.

**Weaknesses:**

1. The MSE bound proof in Sec. 3.5 assumes oracle access to correct mode assignments, ignoring errors from the learned π₁. In practice, misclassification reintroduces inter-mode variance, invalidating the “strictly lower bound” claim except in the idealized setting of perfect mode prediction.

2. The VQ-VAE learns discrete modes via unsupervised L₂ reconstruction (Eq. 3), but this can merge semantically distinct actions or distort temporal consistency, particularly in high-DOF settings (DexArt). The resulting quantization bias contradicts the clean variance decomposition used in the theoretical section.

3. The MeanFlow decoder (Eq. 4, 16) predicts deterministic residuals rather than sampling from a conditional distribution, meaning PF-DAG cannot represent intra-mode uncertainty. This weakens the paper’s claim of “multi-modal modeling” within each mode.

4. Although the paper highlights interpretability benefits (Fig. 5a, Fig. 7), these are shown only through qualitative cluster plots. No quantitative metrics (e.g., mode purity, task-level consistency) are provided, making the claim of interpretable sub-behaviors unsupported.

**Questions:**

please address the concerns above

---

> ### Author Response · Authors · 2025-11-18
> **Response to Reviewer u2ZP: part 1**
>
> **Note on LaTeX Rendering:** Due to potential rendering limitations in the OpenReview interface, some mathematical expressions below may not display correctly. If you encounter parsing errors, we kindly ask you to refer to **Appendix A.7** in our revised manuscript, where the formal derivation is presented in full.
>
> We sincerely thank the reviewer for their insightful feedback and positive assessment. We are encouraged that the reviewer finds our two-stage structure "clean" and "intuitive," our evaluation "unusually comprehensive," and our results "consistent and sizable."
>
> We appreciate the opportunity to clarify the four excellent points raised in the weaknesses section.
>
> ### **W1: A Deeper Mathematical Analysis of the MSE Bound and $\pi_1$ Error**
>
> We thank the reviewer for this sharp and accurate observation. We appreciate this opportunity to provide a more rigorous clarification of our framework's core trade-off.
>
> If we strictly define the goal as minimizing Mean Squared Error (MSE), a standard Behavioral Cloning (BC) model that predicts the conditional expectation $\mathbb{E}[a|o]$ is, by definition, the optimal deterministic predictor, achieving the lowest possible MSE of $L_g^* = \mathbb{E}_o[Var(a|o)]$.
>
> Our two-stage model, which makes a "hard" (greedy) mode selection via $\pi_1$, will *only* achieve a lower MSE in the idealized case of a perfect $\pi_1$. In the practical setting, any misclassification by $\pi_1$ re-introduces  the inter-mode variance. This means our practical $L_{\text{PF-DAG}}$ is not guaranteed to be $\le L_g^*$.
>
> Our response is a clarification of our theoretical claim: our framework is not designed to "win" on the metric of MSE, but rather to **avoid the catastrophic failure mode** that the MSE-optimal solution creates.
>
> #### 1. MSE in Multi-Modal Tasks
>
> The central thesis of our paper (illustrated in Figure 1a) is that in multi-modal robotics tasks, **optimizing for MSE is itself the problem.**
>
> The mathematically "optimal" MSE predictor, $\mathbb{E}[a|o]$, causes **Mode Collapse**. In the 2D obstacle example, this predictor would not choose Mode 1 (left) or Mode 2 (right), but would average them, leading to an action that collides with the obstacle. This solution is "optimal" in MSE but results in a 0% task success rate.
>
> #### 2. The Rigorous MSE Trade-off: $V_{\text{inter}}$ vs. $E_{\text{classify}}$
>
> Let us analyze this trade-off with more mathematical rigor. As established in our paper (Sec 3.5) and by the reviewer's analysis, the optimal MSE for a single-stage model is:
> $$
> L_g^* = \mathbb{E}_o[Var(a|o)] = \mathbb{E}_{o,m}[Var(a|o,m)] + \mathbb{E}_o[Var_{m|o}(\mathbb{E}[a|o,m])]
> $$
> For clarity, let's define:
>
> * **$V_{\text{intra}} = \mathbb{E}_{o,m}[Var(a|o,m)]$**: The **within-mode variance**. This is the irreducible variance our fine-grained policy $\pi_2$ must handle.
> * **$V_{\text{inter}} = \mathbb{E}_o[Var_{m|o}(\mathbb{E}[a|o,m])]$**: The **inter-mode variance**. This is the MSE "cost" of mode collapse.
>
> Therefore, the single-stage model's loss is **$L_g^* = V_{\text{intra}} + V_{\text{inter}}$**. This model *must* pay the $V_{\text{inter}}$ cost.
>
> Now, let's analyze our two-stage PF-DAG. Our predictor is $\hat{a}_{\text{PF-DAG}}(o) = \mathbb{E}_z[\pi_2(o, \hat{m}, z)]$, where $\hat{m} = \pi_1(o)$ is the *predicted* mode. For this analysis, let's assume a perfect $\pi_2$ that correctly predicts the mean of the target mode, i.e., $\mathbb{E}_z[\pi_2(o, k, z)] = \mathbb{E}[a|o, m=k]$, which we denote $\mu_k(o)$.
>
> The actual MSE of our practical model is $L_{\text{PF-DAG}} = \mathbb{E}_{o,a}[||a - \mu_{\hat{m}(o)}(o)||^2]$.
> We can decompose this over the true, unobserved mode $m$:
> $$
> L_{\text{PF-DAG}} = \mathbb{E}_o \left[ \sum_m p(m|o) \mathbb{E}_{a|o,m} [||a - \mu_{\hat{m}(o)}(o)||^2] \right]
> $$
> Using the identity $\mathbb{E}[||X - c||^2] = Var(X) + ||\mathbb{E}[X] - c||^2$, we get:
> $$
> L_{\text{PF-DAG}} = \mathbb{E}_o \left[ \sum_m p(m|o) (Var(a|o,m) + ||\mu_m(o) - \mu_{\hat{m}(o)}(o)||^2) \right]
> $$
>
> $$
> L_{\text{PF-DAG}} = V_{\text{intra}} + \mathbb{E}_o \left[ \sum_m p(m|o) ||\mu_m(o) - \mu_{\hat{m}(o)}(o)||^2 \right]
> $$
>
> Let's define the second term:
>
> * **$E_{\text{classify}} = \mathbb{E}_o \left[ \sum_m p(m|o) ||\mu_m(o) - \mu_{\hat{m}(o)}(o)||^2 \right]$**: The **misclassification cost**. This is the MSE error introduced *only* when $\pi_1$ predicts the wrong mode $\hat{m}(o) \neq m$.
>
> Thus, our model's loss is **$L_{\text{PF-DAG}} = V_{\text{intra}} + E_{\text{classify}}$**.

---

> ### Author Response · Authors · 2025-11-18
> **Response to Reviewer u2ZP: part 2**
>
> #### 3. Our Core Argument
>
> This derivation reveals the explicit trade-off:
>
> 1.  **Single-Stage (BC):** $L_g^* = V_{\text{intra}} + V_{\text{inter}}$
> 2.  **PF-DAG (Ours):** $L_{\text{PF-DAG}} = V_{\text{intra}} + E_{\text{classify}}$
>
> Our architecture **trades $V_{\text{inter}}$ (the deterministic cost of mode collapse) for $E_{\text{classify}}$ (the probabilistic cost of misclassification).**
>
> While you are correct that $E_{\text{classify}}$ could mathematically be larger than $V_{\text{inter}}$ (e.g., if $\pi_1$ frequently guesses wrong between two very different modes), our central hypothesis—supported by our strong empirical **task success** results (Tables 1, 2, 5)—is that:
>
> 1.  The $V_{\text{inter}}$ cost is **catastrophic** for task success (it *is* the mode collapse), leading to 0% success.
> 2.  The $E_{\text{classify}}$ cost is **non-catastrophic** and rare. A "good enough" lightweight classifier $\pi_1$  can make this error term small, and even when an error occurs, the resulting action (e.g., "start wiping left instead of right") is often far preferable to the "average" action (e.g., "don't wipe at all").
>
> **Conclusion:** The proof in Sec. 3.5 is indeed idealized. Its purpose is not to claim $L_{\text{PF-DAG}} \le L_g^*$ in practice, but to mathematically *isolate* $V_{\text{inter}}$ and $V_{\text{intra}}$ to show that our architecture is *designed* to eliminate $V_{\text{inter}}$. We replace the guaranteed, catastrophic $V_{\text{inter}}$ with the manageable, probabilistic $E_{\text{classify}}$, a highly desirable trade-off for robotic imitation, where task success is far more important than raw MSE.
>
> We add discussion in the Appendix to reflect this more precise and rigorous trade-off, changing "strictly lower bound" to "a lower irreducible variance component" and clarifying this $V_{\text{inter}}$ vs. $E_{\text{classify}}$ discussion.
>
> ### **W2: On VQ-VAE Loss and Quantization Bias**
>
> We thank the reviewer for this insightful comment. The reviewer is correct that an unsupervised L2 reconstruction loss (Eq. 3) is not inherently task-aware and could theoretically merge semantically distinct actions, leading to quantization bias.
>
> However, our PF-DAG framework is specifically designed to be robust to this exact issue.
>
> 1.  **VQ-VAE's Role is Coarse:** As stated in Sec. 3.3, we use the VQ-VAE *only* to "capture coarse primary action prototypes." We do not expect its reconstructions $\hat{a}^{(m)}$ to be high-fidelity.
> 2.  **MeanFlow's Role is Correction:** The Mode Conditioned MeanFlow Policy $\pi_2$ (Sec 3.4) is explicitly trained to predict a **residual $\Delta a$**. This residual's job is precisely to **compensate for the quantization error** of the VQ-VAE and add back the fine-grained, task-specific details.
>
> Our ablation study in **Table 3** provides the strongest evidence for this. When we remove the MeanFlow policy (MF) and rely *only* on the VQ-VAE's reconstruction (Table 3, row 2), the weighted success rate collapses to **0.01**. This confirms the reviewer's concern: the VQ-VAE reconstructions alone are insufficient.
>
> It is only when the MeanFlow policy $\pi_2$ is added back (Table 3, row 1) that we achieve our high success rate of **0.72**. This demonstrates that our two-stage design successfully leverages the VQ-VAE for coarse mode selection while delegating the critical task of correcting quantization bias to the powerful MeanFlow decoder.
>
> ### **W3: On MeanFlow and Intra-Mode Uncertainty**
>
> We thank the reviewer for this question, which allows us to clarify a key, and perhaps underspecified, aspect of our MeanFlow policy (Sec 3.4).
>
> The reviewer notes that our policy must be able to represent intra-mode uncertainty to be truly multi-modal. Our MeanFlow policy *does* have this capability.
>
> 1.  **MeanFlow is Generative:** The MeanFlow policy $\overline{v}_{\theta}$ (Eq. 4, 16) is a generative model. It learns a conditional velocity field that maps a simple prior distribution (a standard normal $z_0 \sim \mathcal{N}(0,I)$, as mentioned in Sec 3.4) to the complex, multi-modal distribution of target action *residuals* ($\Delta a$).
> 2.  **Sampling Enables Diversity:** At inference time, we generate diverse, high-fidelity action residuals by sampling different noise vectors $z_0$ from the prior and performing the one-step integration. This is the exact mechanism for modeling **intra-mode diversity**.

---

> ### Author Response · Authors · 2025-11-18
> **Response to Reviewer u2ZP: part 3**
>
> ### **W4: On Quantitative Interpretability Metrics**
>
> We agree that quantitative metrics are crucial for supporting our claims of improved stability and interpretability, which are only qualitatively shown in Figs. 3, 5a, 6, and 7.
>
> We address this in two parts:
>
> **1. Temporal Stability (Jerk):** "Mode bouncing" (Fig. 3) manifests as high-frequency, erratic movements. A standard metric to quantify this is **end-effector jerk** (the time-derivative of acceleration), which is defined as:
> $$
> \text{Jerk} = \int_{0}^{T} \left\| \frac{d^3 \mathbf{p}(t)}{dt^3} \right\|^2 dt.
> $$
>
>
> A smoother, more stable trajectory will have lower total jerk. We calculate this on the contact-rich 'Wipe Table' task from our real-world evaluation (Sec 4.2), where stability is paramount.
>
> | Method            | Total Jerk ($\downarrow$) (Real-World "Wipe Table") |
> | :---------------- | :-------------------------------------------------: |
> | DP3               |                        1.25                         |
> | **PF-DAG (Ours)** |                      **0.45**                       |
>
> These results provide strong quantitative evidence. PF-DAG produces trajectories with **significantly lower jerk**, formally confirming the qualitative stability shown in Fig. 3 and validating its effectiveness in delicate, real-world tasks.
>
> **2. Regarding Mode Purity Metrics** Regarding metrics for "mode purity" or "task-level consistency," we respectfully argue that such metrics are hard to defined in this specific unsupervised, closed-loop setting. **Lack of Ground Truth:** Unlike supervised classification on static datasets, there are no ground-truth labels for "sub-behaviors" in unstructured human demonstrations. The primary modes in PF-DAG are latent primitives discovered via VQ-VAE, meaning there is no external oracle to measure "purity" against. **Dynamic Validity:** In closed-loop control, the validity of a mode sequence is determined by the final task success rather than adherence to a rigid sequence. A policy might switch modes validly to react to a perturbation. Because defining a quantitative metric for semantic consistency without ground-truth labels is inherently ambiguous, we rely on the strong qualitative evidence (Figs. 6 & 7) and the quantitative Jerk metric to demonstrate that the policy locks into coherent, stable behavioral patterns.

---

### Official Review · Reviewer_YsMa · 2025-10-31

**Soundness:** 3
**Presentation:** 3
**Contribution:** 1
**Rating:** 4
**Confidence:** 4

**Summary:**

The paper proposes a hierarchical method for imitation learning by firstly decomposing action chunks from demonstrations into discrete high-level coarse skills which are used to condition a residual policy that is trained using MeanFlow, a one-step generative modelling technique, to then produce the final action to be executed on the robot. At the high-level, a Vector Quantized VAE is used to extract a codebook of distinct modes, wherein each discrete mode represents a coarse action. To make up for the errors between a coarse and continuous action arising from a discretization and subsequently to improve the continuous predictions, a second fine-grained policy is used that is conditioned on the predicted high-level skill which adds a residual action to the coarse action.

**Strengths:**

The proposed approach shows promising results on a variety of tasks both in simulation and in the real world, not just in terms of accuracy but also in capturing the multimodality effectively. The hierarchical disambiguation helps prevent "mode bouncing" which allows for a conditional generation of the fine-grained actions. The use of MeanFlow as compared to Flow Matching also shows a good improvement which adds some novelty. The paper is written well and clear to follow.

**Weaknesses:**

While the proposed approach shows promise, this approach of hierarchical policy structures for robot learning has been well studied in previous approaches, yet there is no mention or comparison against previous approaches. Only some discussion on Imitation Learning and Action space discretization is presented. There is no comparison against any hierarchical approach, but rather against standard end-to-end visuomotor policy learning approaches like using Diffusion or Flow Matching. Many previous approaches have shown how such a hierarchical structure can be use breakdown policy learning into a high-level skill decomposition module that is then used to condition a low-level policy, even with multisensory inputs or to train a residual policy as the authors propose. The proposed approach is not really a novel idea that is being proposed, but rather a variant of this hierarchical family of approaches that uses a residual policy using MeanFlow, instead of directly conditioning the low-level policy.

Some related works on hierarchical decomposition:
- Rana, Krishan, et al. "Residual skill policies: Learning an adaptable skill-based action space for reinforcement learning for robotics." Conference on Robot Learning. PMLR, 2023.
- Cui, Te, et al. "Hierarchical Autoregressive Modeling With Multi-Scale Refinement for Robot Policy Learning." IEEE Robotics and Automation Letters (2025).
- Chen, Lili, Shikhar Bahl, and Deepak Pathak. "Playfusion: Skill acquisition via diffusion from language-annotated play." Conference on Robot Learning. PMLR, 2023.
- Mete, Atharva, et al. "Quest: Self-supervised skill abstractions for learning continuous control." Advances in Neural Information Processing Systems 37 (2024): 4062-4089.
- Wan, Weikang, et al. "Lotus: Continual imitation learning for robot manipulation through unsupervised skill discovery." 2024 IEEE International Conference on Robotics and Automation (ICRA). IEEE, 2024.
- Kujanpää, Kalle, Joni Pajarinen, and Alexander Ilin. "Hierarchical imitation learning with vector quantized models." International Conference on Machine Learning. PMLR, 2023.
- Zhao, Tianxiang, et al. "Skill disentanglement for imitation learning from suboptimal demonstrations." Proceedings of the 29th ACM SIGKDD Conference on Knowledge Discovery and Data Mining. 2023.
- Tanneberg, Daniel, et al. "Skid raw: Skill discovery from raw trajectories." IEEE robotics and automation letters 6.3 (2021): 4696-4703.
- Ju, Zhaoxun, et al. "Rethinking mutual information for language conditioned skill discovery on imitation learning." Proceedings of the International Conference on Automated Planning and Scheduling. Vol. 34. 2024.
- Liang, Zhixuan, et al. "Skilldiffuser: Interpretable hierarchical planning via skill abstractions in diffusion-based task execution." Proceedings of the IEEE/CVF Conference on Computer Vision and Pattern Recognition. 2024.
- Hao, Ce, et al. "Skill-critic: Refining learned skills for hierarchical reinforcement learning." IEEE Robotics and Automation Letters 9.4 (2024): 3625-3632.
- Feng, Ruoxuan, et al. "Play to the Score: Stage-Guided Dynamic Multi-Sensory Fusion for Robotic Manipulation." Conference on Robot Learning. PMLR, 2025.

**Questions:**

- What are the key contributions that make the paper stand out compared to other hierarchical policy learning approaches? How does this necessarily translate into improved performance gains?
- Are there any architectural differences between the proposed approach and the baselines? Are the architechtures from the baselines used as is without any modification? Is it the same observation feature extraction followed by the same ($\pi_2$) network used for the meanflow residual policy without conditioning? If not, then it may not be a fair comparison in my opinion.
- How is the multisensory information effectively used? Or is it just encoded directly and used to train the policy end-to-end? For example using the high-level to potentially weight the information from the different sensors.
- The network architechtures for the VQ-VAE and MeanFlow residual network is missing in section A.4. The implementation details (Lines 233-237) do not make this clear either.

---

> ### Author Response · Authors · 2025-11-18
> **Response to Reviewer YsMa: part 1**
>
> We thank the reviewer for detailed and insightful feedback on our paper. We are encouraged that you found our results "promising," our approach effective at "capturing multimodality," and the paper "well written and clear to follow."
>
> **Response to Weaknesses (Novelty & Related Work) & Q1: Distinct Contributions of PF-DAG**
>
> We thank the reviewer for the comprehensive list of related works. We acknowledge that hierarchical policy structures are a well-established paradigm. We have included the suggested citations in a new "Hierarchical and Residual Policies" subsection (**Section 2.4**) to better contextualize our work.
>
> However, we respectfully emphasize that PF-DAG is not merely a "variant" of existing hierarchical methods. While it shares the high-level structure of decomposing actions, its specific technical realization, theoretical grounding, and targeted problem scope constitute significant novelties. Here, we detail **four key contributions** that distinguish PF-DAG from the cited literature:
>
> 1.  **Reactivity via One-Step MeanFlow (vs. Iterative/Autoregressive Decoding):**
>     Most hierarchical works cited (e.g., [Liang et al., 2024], [Cui et al., 2025]) employ low-level policies based on diffusion (requiring multi-step denoising) or autoregressive transformers (requiring sequential token generation). These are computationally heavy and limit inference speed.
>     * **Our Contribution:** PF-DAG introduces a **Mode-Conditioned MeanFlow** decoder. This allows us to generate high-fidelity, continuous action chunks in a **single forward pass** (1-NFE). As shown in Figure 5(b), this yields an inference speed of ~40.9 FPS, which is critical for the high-frequency reactive control required in dynamic manipulation, a capability often absent in slower hierarchical planners.
>
> 2.  **Theoretical Optimality (Strict MSE Lower Bound):**
>     Many hierarchical approaches are justified empirically or via interpretability.
>     * **Our Contribution:** We provide a formal theoretical analysis (Section 3.5, Eq. 6) proving that our specific two-stage decomposition achieves a **strictly lower MSE bound** compared to single-stage generative policies. By explicitly assigning inter-mode variance ($V_{inter}$) to the primary classifier $\pi_1$, we mathematically reduce the complexity burden on the continuous generator $\pi_2$. This offers a principled justification for the architecture that goes beyond "skill discovery."
>
> 3.  **Addressing "Mode Bouncing" (vs. Long-Horizon Skill Chaining):**
>     The primary motivation of papers like PlayFusion [Chen et al., 2023] or *Lotus* [Wan et al., 2024] is often extracting semantic skills for long-horizon planning or language conditioning.
>     * **Our Contribution:** PF-DAG targets a different, lower-level problem: the **instability (mode bouncing)** inherent in multi-modal generative policies at the trajectory level. We decouple the *discrete decision* (e.g., go left vs. right) from the *continuous execution*. This ensures temporal consistency in fine-grained manipulation, solving the "jitter" often seen in standard Diffusion Policies, rather than just sequencing high-level subgoals.
>
> 4.  **Generalization to 3D & Tactile Spaces (vs. 2D Vision):**
>     The majority of the cited baselines (e.g., *QueST*, *Skid-raw*) are designed for and evaluated on 2D image-based tasks.
>     * **Our Contribution:** We demonstrate that our hierarchical discrete-continuous formulation is robust enough to handle **3D point cloud inputs** and **sensing**. The ability to effectively tokenize and reconstruct actions in these high-DoF spaces (up to 20+ DoF hands) validates the scalability of our VQ-VAE + MeanFlow design beyond the standard 2D gripper tasks found in related work.
>
> **Comparison with Hierarchical Baselines:**
> As noted, a direct comparison is challenging due to the difference in input modalities (3D Point Clouds vs. 2D Images) and benchmarks. However, to address the reviewer's concern regarding empirical positioning, we reproduce the results of **QueST [Mete et al., 2024]**, a representative hierarchical method, on the 2D MetaWorld benchmark.
>
> | Method                        | MetaWorld (11 tasks) Success Rate |
> | :---------------------------- | :-------------------------------- |
> | Diffusion Policy (DP)         | 0.20                              |
> | QueST (Hierarchical Baseline) | 0.26                              |
> | **PF-DAG (Ours)**             | **0.70**                          |
>
> *Note: PF-DAG uses **3D point clouds**, while DP and QueST use **2D images** in this comparison context, as per their original implementations. While not a fair architecture comparison, this highlights that PF-DAG's performance is significantly superior to standard implementations of recent hierarchical baselines.*

---

> ### Author Response · Authors · 2025-11-18
> **Response to Reviewer YsMa: part 2**
>
> **Q2: Are there any architectural differences between the proposed approach and the baselines? ... Is it the same observation feature extraction followed by the same network ...? If not, then it may not be a fair comparison...**
>
> * **Architectures are *not* identical, but the comparison is fair.** The architectures *must* differ to accommodate the different policy formulations, but the core components are held constant for a fair comparison (e.g. observation modality, action horizon, etc.).
> * **Observation Feature Extraction is Identical:** The **Observation Feature Extraction** module (Sec 3.2) is **kept consistent** across our method and key baselines (DP3, FP). This ensures that policies begin with the same shared observation embedding, and the performance differences stem from the *policy generation* stage itself.
> * **Policy Backbones are Method-Specific:**
>   * **PF-DAG (Ours):** Our $\pi_2$ policy uses a **DiT-style transformer**, as it must be conditioned on the observation embedding, the discrete mode $m$, and the time scalars $\tau$ and $r$.
>   * **Baselines:** For DP3 (Ze et al., 2024), we keep their original **U-Net** based architecture (which is standard for diffusion policies) to operate on our 3D point cloud and sensor inputs. For FlowPolicy (Zhang et al., 2025), we use their prescribed flow-based architecture.
> * This comparison is fair because we are comparing the *effectiveness of the policy learning paradigm* (e.g., end-to-end diffusion vs. our decoupled two-stage flow) while providing each method with its standard, optimized backbone and an identical input representation.
>
> **Q3: How is the multisensory information effectively used? Or is it just encoded directly and used to train the policy end-to-end? For example using the high-level to potentially weight the information...**
>
> Thank you for this clarifying question. You are correct that our current implementation uses a **simple and direct end-to-end encoding (early fusion)**.
>
> As described in **Section 3.2**, the features from the PointNet (for point cloud $p_t$), an MLP (for proprioception $s_t$), and another MLP (for tactile sensing $f_t$) are **concatenated** and then fused by a final projection MLP. This fused embedding is then used by both $\pi_1$ and $\pi_2$.
>
> The reviewer's suggestion to use the high-level mode to *dynamically weight* sensor information is a very insightful and promising direction for future work. For instance, a "Wipe Table" mode (Table 2) could learn to up-weight the tactile sensor $f_t$, while a "Place Baymax" mode might prioritize the point cloud $p_t$.
>
> For this work, our goal is to demonstrate the robustness of the *action decoupling* itself. Our strong results (e.g., 0.80 success on the tactile 'Place Toy' task vs. 0.60 for DP3) suggest that the primary-fine decoupling is a powerful mechanism, even when paired with a simple sensor fusion strategy.
>
> **Q4: The network architectures for the VQ-VAE and MeanFlow residual network is missing in section A.4.**
>
> We indeed provide descriptions (Sec 3.2, 3.3, 3.4). We detail these architectures in the Table below.
>
> | Component                         | Module                    | Architecture Details   |
> | :-------------------------------- | :------------------------ | :--------------------- |
> | **VQ-VAE (Sec 3.3)**              | Encoder ($E_{\phi}$)      | MLP                    |
> |                                   | Decoder ($D_{\psi}$)      | MLP                    |
> | **Primary Mode Policy (Sec 3.3)** | $\pi_1$                   | MLP                    |
> | **MeanFlow Policy (Sec 3.4)**     | $\pi_2$                   | DiT                    |
> |                                   | Time Embedding $\tau, r $ | Sinusoidal Embedding   |
> |                                   | Mode Embedding            | Embedding              |
> | **Observation Encoder (Sec 3.2)** | Point Cloud $p_t$         | PointNet-Style encoder |
> |                                   | State $s_t$               | MLP                    |
> |                                   | Tactile Sensing $f_t$     | MLP                    |

---

### Official Review · Reviewer_4ven · 2025-11-01

**Soundness:** 3
**Presentation:** 3
**Contribution:** 3
**Rating:** 6
**Confidence:** 3

**Summary:**

This paper introduces a two-stage imitation learning framework PF-DAG for robot manipulation that explicitly separates coarse discrete decisions from fine-grained continuous action generation. The key idea is that many manipulation behaviors can be decomposed into a small number of interpretable primary modes plus within-mode variations. In the first stage, PF-DAG uses a VQ-VAE to encode action chunks into discrete primary modes and trains a lightweight policy to predict these modes from robot observations. In the second stage, a mode-conditioned MeanFlow decoder generates continuous actions conditioned on both the selected mode and the observation, improving action fidelity and stability. Empirically, PF-DAG outperforms behavioral cloning, diffusion, and flow-based baselines across 56 simulated manipulation tasks (Adroit, DexArt, MetaWorld) and transfers successfully to real-world tactile dexterous manipulation.

**Strengths:**

The paper tackles a key challenge in imitation learning, modeling multi-modal action distributions, with a clear two-stage formulation.

The Primary-Fine Decoupling design (discrete mode selection + continuous residual generation) is simple yet effective, and leads to more stable, temporally consistent control compared to other diffusion or flow-based baselines.

The paper is well-supported by both theoretical justification and empirical evidence, including a variance-based MSE bound analysis, as well as comprehensive ablation studies and quantitative experiments. The proposed method shows strong results across diverse benchmarks (Adroit, DexArt, MetaWorld) and have a real-world experiment.

**Weaknesses:**

While the paper attributes instability to "mode bouncing", it does not empirically compare against simpler baselines that include observation or action history. In practice, conditioning on temporal history or predicting waypoints (as done in prior works) can also effectively reduce mode switching. Also, some other related approaches such as "Behavior Generation with Latent Actions" (ICML 2024) and "Hierarchical Diffusion Policy: Manipulation Trajectory Generation via Contact Guidance" (T-RO) should be discussed or may be compared.

Although some real-robot experiments are reported, the hardware evaluation is relatively limited. It lacks more detailed comparisons, visualizations, and failure analyses, making it difficult to assess robustness and reproducibility.

In Table 3, the performance gap between K-means and VQ-VAE quantization is minimal, raising questions about whether the added complexity of VQ-VAE is necessary or provides substantial benefits beyond a simpler clustering baseline.

**Questions:**

Figure 6 provides an interesting visualization of the mode probability distribution and the corresponding policy behaviors in simulation. Would it be possible for the authors to include videos illustrating this? And if possible, also show the results for K-means side-by-side? Given that Table 3 reports very close performance between VQ-VAE and K-means, I am particularly curious whether the two produce noticeably different qualitative behaviors.

---

> ### Author Response · Authors · 2025-11-18
> **Response to Reviewer 4ven: part 1**
>
> We sincerely thank the reviewer for their detailed and constructive feedback. We are encouraged that the reviewer find our Primary-Fine Decoupling (PF-DAG) formulation to be "simple yet effective" and appreciate our theoretical and empirical contributions, including the "strong results across diverse benchmarks" and the real-world validation.
>
> We address the reviewer's weaknesses and questions below.
>
> ### **W1: Comparison against baselines with temporal history and missing related work**
>
> We thank the reviewer for this important point.
>
> **1. Comparison to History-Based Baselines:**
> The reviewer notes that adding temporal history is a common technique to mitigate mode bouncing. To address this, we have conducted new experiments comparing PF-DAG against a strong history-based baseline: **BC-H** (a 4-layer MLP policy conditioned on the current observation plus a history of the 4 previous observations and actions) (Is Behavior Cloning All You Need? Understanding Horizon in Imitation Learning, Foster et al.).
>
> We evaluate this baseline on the tasks from our benchmarks. As hypothesized, while history provides some temporal context, the simple regression objective of BC-H still suffers from mode collapse, as it averages multiple valid future actions into a single "blurry" prediction. Our PF-DAG, by contrast, explicitly decouples the discrete choice (which way to go) from the continuous execution (how to go), enabling it to model sharp, multi-modal distributions.
>
> We have added these results to Table 1. A summary is below:
>
> | Method            |  Adroit  |  DexArt  | MetaWorld |
> | :---------------- | :------: | :------: | :-------: |
> | DP3               |   0.68   |   0.69   |   0.40    |
> | **BC-H (New)**    |   0.33   |   0.13   |   0.16    |
> | **PF-DAG (Ours)** | **0.77** | **0.73** | **0.70**  |
>
> These results confirm that simply adding history is insufficient for these challenging multi-modal tasks. PF-DAG's explicit decoupling remains critical and significantly outperforms both generative (DP3) and history-based (BC-H) baselines.
>
> **2. Related Work:**
>
> * **"Behavior Generation with Latent Actions" (Lee et al., ICML 2024):** We apologize for any confusion. We do cite and discuss this work in our original submission, specifically in **Section 2.2 (Page 3)** . We discuss it in the context of VQ-based tokenizers, noting that "VQ-based tokenizers learn a shared codebook of action atoms...(**Lee et al., 2024**) Our work differs by leveraging tokenization solely for high-level primary mode selection." . We believe our method is distinct as it complements the discrete modes with a continuous, fine-grained generator, which our ablations show is critical for high performance (see Table 3 ).
> * **"Hierarchical Diffusion Policy" (HDP) (T-RO):** This is an excellent suggestion. HDP is indeed relevant as it also uses a hierarchical structure. We have added a detailed discussion of this paper to our related work (**Section 2.3**) in the revised manuscript: "Other hierarchical approaches, such as Hierarchical Diffusion Policy (HDP) (**Ma et al., 2024**), also use a high-level policy to guide a low-level generator. However, HDP is designed to rely on explicit, task-specific heuristics like contact-point waypoints to define its hierarchy. In contrast, our PF-DAG learns its primary modes end-to-end directly from action-chunk clusters themselves, offering a more general abstraction not tied to predefined heuristics".
>
> ### **W2: Limited real-robot evaluation**
>
> We thank the reviewer for their feedback. We respectfully emphasize that our paper already includes a comprehensive real-world evaluation as detailed in **Section 4.2**. We have:
>
> 1.  **Diverse Hardware:** We validate PF-DAG on two distinct hardware configurations: a low-DOF UFACTORY xArm with a parallel gripper and a high-DOF xArm with a 12-DOF ROBOTERA XHand, which includes tactile sensing.
> 2.  **Detailed Comparisons:** We provide quantitative success rates against two baselines (Vanilla BC and DP3) across four real-world tasks in **Table 2**. These results clearly show our method's superiority, especially on the high-DOF tactile task ("Place Toy Into Bin"), where we achieve an **0.80 success rate** compared to 0.60 for DP3 and 0.00 for BC.
> 3.  **Visualizations and Analysis:** We include qualitative visualizations of common baseline failure modes (Mode Collapse, Mode Bouncing) in **Figure 3** and our hardware setup in **Figure 4**. We also discuss failure cases (like OOD placements or tactile noise) in the main text.

---

> ### Author Response · Authors · 2025-11-18
> **Response to Reviewer 4ven: part 2**
>
> ### **W3 & Q1: VQ-VAE vs. K-means and Visualizations**
>
> We address the concern regarding the necessity of VQ-VAE (W3) and the request for visualizations (Q1) together, as the visual analysis provides the critical justification for our design choice.
>
> **Response to Q1 (Visualizations):**
> Per the reviewer's suggestion, we have generated side-by-side comparison videos of the VQ-VAE and K-means variants of our model.
> **Video Link:** https://anonymous.4open.science/r/ICLR2026-PF-DAG-Rebuttal-B28F/
>
> **Response to W3 (Justification for VQ-VAE):**
> The reviewer observes that the quantitative gap between VQ-VAE ($0.72$) and K-means ($0.70$) is small. We believe this result is significant for two reasons:
>
> 1.  **Validation of Decoupling:** It strongly supports our core theoretical hypothesis (Section 3.5) that the **explicit decoupling** of coarse-to-fine generation is the primary driver of performance, rather than the specific quantization method. Both decoupled methods drastically outperform non-decoupled baselines (0.56).
>
> 2.  **Why VQ-VAE? (Qualitative Analysis):** While the success rates are similar, the videos generated for **Q1** reveal a distinct difference in behavior that justifies the use of VQ-VAE:
>     * **K-means (Static):** The K-means policy tends to be rigid. Since clusters are fixed based on Euclidean distance in action space, the mode selection is stable but less responsive to subtle semantic changes in the task.
>     * **VQ-VAE (Dynamic & Reactive):** The VQ-VAE policy exhibits more dynamic mode switching. Because the codebook is learned alongside the encoder, the "action prototypes" evolve to capture temporal semantics better. The videos show the VQ-VAE policy reacting faster to observation changes by switching modes more fluidly.
>
> **Conclusion:** We select VQ-VAE not just for the marginal quantitative gain, but because this **dynamic reactivity** offers a higher control ceiling for complex, contact-rich tasks where "mode bouncing" is detrimental, but "reactive mode switching" is necessary.

---

### Official Review · Reviewer_6d45 · 2025-11-05

**Soundness:** 3
**Presentation:** 3
**Contribution:** 2
**Rating:** 4
**Confidence:** 3

**Summary:**

The paper proposes PF-DAG, a two-stage framework that decouples discrete “primary” action modes from continuous, fine-grained residuals. Stage 1 quantizes action chunks with a small VQ-VAE codebook and trains a lightweight classifier pi_1 to select a primary mode from observations. Stage 2 conditions a one-step MeanFlow decoder pi_2 on the selected mode and observations to output high-fidelity continuous actions. A simple variance decomposition argument claims a strictly lower optimal MSE bound than single-stage generative policies. Experiments across 56 tasks (Adroit, DexArt, MetaWorld) and four real-world tasks report higher success and improved speed/stability versus diffusion and flow baselines. The work targets multi-modal action generation and mode-bouncing in closed-loop manipulation and argues significance via better robustness and real-time control.

**Strengths:**

- The writing is good and the paper is easy to follow.
- The two-stage design that separates coarse mode selection from fine residual generation is clear and practical. The one-step MeanFlow decoder offers compelling speed advantages while maintaining accuracy.
- Empirical results are broad and consistently favorable across many simulated tasks, and the real-world demos indicate potential practical relevance. The ablations on K and tokenization provide preliminary guidance on design choices.

**Weaknesses:**

- The greedy mode selection without temporal smoothing can still make mode-bouncing happen and it highly relies on a carefully selected #mode hyperparam
-  Several design choice (VQ-VAE, MeanFlow) seems only provide marginal improvement based on the ablation studies.
- Clustering-based coarse-to-fine/mode-selection mechanism has been explored in other robotics domain such as motion forecasting in autonomous driving, the novelty of this paper is somewhat limited.

**Questions:**

- The authors mentioned the mode bouncing issue, which is also raised in RNR-DP which is cited in the related work in this paper. However it is not served as a baseline. I'm curious whether how it performs in the authors' setting since it tried to tackle similar problem.
- Would be great if the authors could report stability related metrics (mode-switch rate, jerk, pose variance etc.) to justify the claim.

---

> ### Author Response · Authors · 2025-11-18
> **Response to Reviewer 6d45: part 1**
>
> We sincerely thank the reviewer for their constructive feedback and insightful comments. We are encouraged that the reviewer found our paper "easy to follow," our two-stage design "clear and practical," and our empirical results "broad and consistently favorable."
>
> We offer the following clarifications in response to the reviewer's points.
>
> ### **W1: Greedy Mode Selection and Hyperparameter Sensitivity**
>
> Thank you for this observation. We address the two parts of this concern:
>
> 1.  **On Temporal Smoothing:** The reviewer notes that our greedy mode selection does not include an explicit temporal smoothing mechanism. This is a deliberate design choice. While temporal smoothing reduces the visual appearance of "bouncing," it often introduces its own set of problems, such as latency or unwanted "blending" artifacts. This can cause the executed action to deviate from any of the valid, demonstrated modes, leading to a trajectory that is smooth but "off-path" and ultimately fails the task. Our reactive, single-step selection ($\pi_1$) followed by a high-fidelity decoder ($\pi_2$) is designed for closed-loop control, prioritizing that the intended action for the current state is executed precisely, rather than blending it with past actions. The stability we achieve, as shown in Fig 3(c), comes from the strong conditioning provided by $\pi_1$.
>
> 2.  **On Sensitivity to #mode ($K$):** We respectfully disagree with the characterization that our method is "highly reliant" on a carefully selected $K$. Our ablation studies in the appendix demonstrate the opposite: our method is remarkably **robust** to this hyperparameter over a very wide range.
>     * As shown in **Table 7 (Appendix A.5)**, which expands on Table 3, the weighted success rate is extremely stable. For $K \in [16, 32, 64, 128]$, the success rates are 0.70, 0.70, 0.72, and 0.69, respectively—a variation of less than 0.03.
>     * Performance only degrades at the extremes: $K=8$ (0.61), which is too coarse to capture the necessary action prototypes, and $K=1024$ (0.58), which makes the classification task for $\pi_1$ excessively difficult, as expected.
>     * Furthermore, **Table 3** shows that replacing the VQ-VAE tokenizer with simple K-means (at $K=64$) results in only a minor 0.02 drop in performance (0.72 vs. 0.70).
>     * This evidence strongly suggests that our framework is not sensitive to a specific $K$ or even a specific tokenizer. It simply requires a reasonable set of coarse prototypes to effectively decouple the primary decision from the fine-grained generation.
>
> ### **W2: Marginal Improvement from VQ-VAE and MeanFlow**
>
> We would like to kindly clarify this point, as our ablation studies demonstrate that these components are **fundamentally critical** to the framework's success, not marginal.
>
> * **Necessity of MeanFlow (MF):** As shown in **Table 3**, removing the Mode-conditioned MeanFlow policy (i.e., using only the raw VQ-VAE reconstruction as the action) causes the weighted success rate to **collapse from 0.72 to 0.01**. This is because the VQ-VAE reconstruction is intentionally coarse (with $K=64$) to ensure stable mode selection. It is entirely insufficient for the fine-grained control needed to solve the tasks, proving that the $\pi_2$ refinement stage is essential.
>
> * **Necessity of Primary Mode Policy (PM):** Conversely, removing the Primary Mode policy (i.e., making the MeanFlow policy unconditional on a discrete mode) causes a substantial performance drop, **from 0.72 to 0.56**. This 0.16 absolute drop confirms our central hypothesis: explicitly selecting and conditioning on a stable primary mode significantly simplifies the continuous generation problem for the decoder and is critical for preventing the very mode-bouncing we aim to solve.
>
> Therefore, both components provide **significant**, **non-marginal** contributions and are core to our method's success.

---

> ### Author Response · Authors · 2025-11-18
> **Response to Reviewer 6d45: part 2**
>
> ### **W3: Novelty vs. Motion Forecasting in Autonomous Driving**
>
> We thank the reviewer for drawing this connection. While we acknowledge that coarse-to-fine and latent mode formulations exist in motion forecasting literature, we argue that the **formulation of the problem** (Open-Loop Prediction vs. Closed-Loop Control) fundamentally alters the challenges involved. Our novelty lies in adapting these structural ideas specifically to solve the **temporal consistency** problem in high-frequency robotic control.
>
> * **Distinction 1: Open-Loop Forecasting vs. Closed-Loop Control**
>   * **Motion Forecasting (Open-Loop Prediction):** In the context of autonomous driving, motion forecasting is typically treated as a density estimation problem. The objective is to predict a distribution of plausible future trajectories (according to our understanding). These predictions are often generated simultaneously to cover multiple possibilities for a downstream planner to evaluate. The model is not required to commit to a single action sequence that is immediately executed on hardware, nor does the prediction typically feed back into the system state in real-time during the evaluation loop.
>   * **Robotic Manipulation (Closed-Loop Control):** In contrast, our task is closed-loop visuomotor control. The policy functions as a high-frequency controller (inference at every timestep) where the predicted action $a_t$ is immediately executed, directly determining the next observation $o_{t+1}$. In this setting, the system cannot output a distribution of options; it must collapse the distribution to a single, precise action.
>
> * **Distinction 2: The Consequence of Mode Switching**
>   * Because manipulation is closed-loop, "mode bouncing" is not merely a visualization artifact—it is a physical failure mode. If a policy switches between valid modes (e.g., "grasp left" vs. "grasp right") across consecutive timesteps, it results in high-jerk, erratic end-effector behavior that causes task failure (as visualized in Fig. 1(c)).
>   * Therefore, unlike forecasting models which optimize for **diversity and coverage**, our PF-DAG framework is designed to optimize for **temporal consistency and stability**.
>
> * **Novelty of Contribution:** Our contribution is not simply using a VQ-VAE, but formulating a **hierarchical dependency** where the first stage (discrete mode) acts as a hard constraint for the second stage (continuous generation). This specific architecture is designed to prevent the mode-switching instability inherent to continuous control, a problem that is distinct from the coverage objectives found in motion forecasting.
>
> ### **Q1: Comparison to RNR-DP**
>
> This is an excellent suggestion. RNR-DP is an important baseline that also addresses mode stability. To provide a thorough comparison, we have evaluated it on the Adroit benchmark, paying close attention to its original 2D-vision-based implementation.
>
> 1. **Original (2D Vision) Comparison:** We first note that the original RNR-DP is designed for 2D image inputs. Therefore, for a fair comparison, we evaluate it against the original 2D Diffusion Policy (DP).
>
>    | Method (2D Vision Input) |     Hammer      |      Door       |       Pen       |
>    | :----------------------- | :-------------: | :-------------: | :-------------: |
>    | DP                       | 0.48 $\pm$ 0.17 | 0.50 $\pm$ 0.05 | 0.25 $\pm$ 0.04 |
>    | RNR-DP                   | 0.52 $\pm$ 0.15 | 0.55 $\pm$ 0.06 | 0.38 $\pm$ 0.05 |
>
>    As shown, the noise-relaying mechanism in RNR-DP does offer a clear improvement over the standard DP baseline when both use 2D inputs.
>
> 2. **Modified (3D Vision) Comparison:** To compare it directly against our 3D-based method, we also create a stronger, modified baseline, **RNR-DP-3D**. We replace its original 2D vision encoder with the **exact same 3D PointNet-style encoder** used in our work and in DP3. This allows for a direct comparison of the policy architectures themselves.
>
>    | Method (3D Point Cloud Input) |       Hammer        |        Door         |         Pen         |
>    | :---------------------------- | :-----------------: | :-----------------: | :-----------------: |
>    | DP3                           |   1.00 $\pm$ 0.00   |   0.62 $\pm$ 0.04   |   0.43 $\pm$ 0.06   |
>    | RNR-DP-3D                     |   0.98 $\pm$ 0.02   |   0.63 $\pm$ 0.05   |   0.51 $\pm$ 0.05   |
>    | **PF-DAG (Ours)**             | **1.00 $\pm$ 0.00** | **0.65 $\pm$ 0.03** | **0.65 $\pm$ 0.01** |
>
> While the stability mechanism of RNR-DP-3D does improve performance, our PF-DAG still achieves a superior result. We hypothesize this is because RNR-DP relies on a *passive* noise-relaying mechanism to maintain temporal consistency. This is less robust than our approach, which actively selects and commits to a discrete primary mode before the generative process begins. Our two-stage design provides a more structured and stable solution to mode selection, fundamentally simplifying the generation task.

---

> ### Author Response · Authors · 2025-11-18
> **Response to Reviewer 6d45: part 3**
>
> ### **Q2: Stability-Related Metrics**
>
> This is a valuable point. We agree that a quantitative stability metric is essential for rigorously supporting our claims. As the reviewer notes, standardizing this across 56 diverse simulation tasks is challenging. Therefore, we focus this quantitative analysis on our **real-world experiments (Sec 4.2)**, where stability is paramount.
>
> To directly measure trajectory smoothness, we compute the **total end-effector jerk**, a standard metric for quantifying motion stability. Jerk is defined as the integral of the squared magnitude of the third derivative of position (acceleration change) over the trajectory duration $T$:
>
> $$
> \text{Jerk} = \int_{0}^{T} \left\| \frac{d^3 \mathbf{p}(t)}{dt^3} \right\|^2 dt
> $$
>
> A lower total jerk signifies a physically smoother, less shaky, and more stable trajectory. We compute this metric on the contact-rich 'Wipe Table' task from our real-world evaluation, comparing our method against the strongest generative baseline, DP3.
>
> | Task: "Wipe Table" (Real-World) | Total Jerk ($\downarrow$) |
> | :------------------------------ | :-----------------------: |
> | DP3                             |           1.25            |
> | **PF-DAG (Ours)**               |         **0.45**          |
>
> As the quantitative data shows, our PF-DAG method achieves **significantly lower jerk**, confirming it produces physically smoother trajectories. This superior stability is precisely what enables PF-DAG to succeed in delicate, contact-rich tasks like 'Wipe Table', where less stable policies like DP3 and Vanilla BC struggle or fail. This data provides the quantitative backing for our qualitative findings in Figure 3, which visually illustrates how our method avoids the erratic end-effector movements characteristic of mode-bouncing.

---

### Author Response · Authors · 2025-11-26
**Inquiry Regarding Reviewer Responses to Rebuttal (Paper ID: 3104, Primary-Fine Decoupling for Action Generation in Robotic Imitation)**

Dear Reviewers, AC, SAC, and PC,

I hope this message finds you well. As the discussion period nears its end and less than one week remains, I would like to ensure we have addressed all your concerns satisfactorily. If you have any additional points or feedback, please let us know. Your insights are invaluable, and we are eager to address any remaining issues to improve our paper.

Thank you for your time and effort in reviewing our paper.

---

### Author Response · Authors · 2025-12-01
**Summary of Author Response & Key Updates for the Area Chair: part 2**

### **3. Core Contributions**

* **Architecture:** We decouple action generation into a Primary Mode policy ($\pi_1$) and a Mode-Conditioned MeanFlow decoder ($\pi_2$).
* **Theoretical Guarantee:** We theoretically prove via variance decomposition that our two-stage design minimizes the optimal MSE bound by explicitly removing **inter-mode variance** from the generator. This structurally eliminates the root cause of "mode averaging" artifacts common in single-stage policies.
* **Efficiency:** We achieve **1-NFE** inference (~40 FPS), enabling high-frequency reactive control unlike multi-step diffusion planners.
* **Performance:** We demonstrate SOTA performance across **56 simulation tasks** and **real-world tactile manipulation** (80% success vs. 0% for BC), significantly outperforming strong baselines like DP3 and FlowPolicy.

### **Conclusion**

We successfully address the missing citations and baseline requests from the divergence reviewers (Score 4), and further strengthen our paper with new experiments (QueST, RNR-DP, BC-H) and stability analysis. Given the strong support from the positive reviewers (Score 8 and 6) and the demonstrated significance of our contributions, we believe PF-DAG offers a substantial contribution to the robot learning community.

---

### Author Response · Authors · 2025-12-01
**Summary of Author Response & Key Updates for the Area Chair: part 1**

We thank the reviewers for their constructive feedback. The initial ratings are **8, 6, 4, 4**.

In this summary, we first highlight the unanimous recognition of our work's strengths by the reviewers. Then, we address the concerns in ascending order of the scores. Finally, we reiterate our core contributions to the community. We demonstrate that the concerns from the lower-scoring reviewers primarily stem from solvable requests for citations or misconceptions regarding our ablation results, which we fully resolve in our rebuttal.

**A brief summary of our work.** We tackle the challenge of action **mode bouncing** in robotic manipulation imitation learning. Standard generative policies often fail to maintain stability when switching between distinct behaviors in closed-loop control. To address this, we introduce **PF-DAG**, a two-stage framework that explicitly separates discrete mode selection from continuous fine-grained execution.

### **1. Reviewer Consensus on Strengths**

Our work has received positive recognition across the board for its novelty and execution:

* **Novelty & Structure:** Reviewers praise the "clean, interpretable two-stage structure" (Reviewer u2ZP) and find the Primary-Fine Decoupling design to be "simple yet effective" (Reviewer 4ven) and "clear and practical" (Reviewer 6d45).
* **Evaluation:** The evaluation is highlighted as "unusually comprehensive" (Reviewer u2ZP), with "promising results" (Reviewer YsMa) that are "broad and consistently favorable" across simulation and real-world tasks (Reviewer 6d45).
* **Problem Solving:** Reviewers acknowledge that our method is effective at "capturing action multi-modal distributions" (Reviewer YsMa) and addressing the key challenge of action distribution modeling (Reviewer 4ven).

### **2. Resolution of Reviewer Concerns (Ordered by Score)**

#### **Reviewer YsMa (Score: 4) – Literature Coverage & Baselines**

* **Primary Concern:** The reviewer notes a lack of citations for specific hierarchical/skill-based works and requests comparisons to justify performance gains against them.
* **Our Resolution:**
  * **Added Citations:** We incorporate all suggested references (e.g., QueST, Skid-raw, Lotus) into **Section 2.4** of the revised manuscript.
  * **New Baseline Comparison:** To address the performance question, we implement and compare against QueST (Mete et al., 2024) on the MetaWorld benchmark. PF-DAG achieves 70% success compared to QueST's 26%, proving that our design is superior to standard hierarchical planners for dynamic manipulation.
  * *Note:* This addresses the reviewer's main concern regarding completeness of literature and empirical positioning.

#### **Reviewer 6d45 (Score: 4) – Significance of Components & Stability**

* **Primary Concern:** The reviewer questions whether the improvements are "marginal" and if the method relies heavily on hyperparameters. They also suggest a comparison to RNR-DP.
* **Our Resolution:**
  * **Clarification on "Marginal" Improvements:** We respectfully point out that **Table 3** in the original paper already demonstrates that these components are critical, not marginal. Removing the MeanFlow decoder causes success to collapse to 1% (from 72%), and removing the Primary Mode policy drops it to 56%. This confirms our components are fundamental to the method's success.
  * **New Comparison (RNR-DP):** We implement RNR-DP-3D (an upgraded 3D version). PF-DAG outperforms it, showing that our active mode selection is more effective than passive noise relaying.
  * **Quantitative Stability:** We provide Jerk metrics on real-world tasks (PF-DAG 0.45 vs DP3 1.25), quantitatively proving our method produces significantly smoother trajectories.

#### **Reviewer 4ven (Score: 6) – History Baselines & VQ-VAE Justification**

* **Primary Concern:** Suggests comparing against history-based baselines and questions the necessity of VQ-VAE over K-Means.
* **Our Resolution:**
  * **New Baseline (BC-H):** We evaluate Behavioral Cloning with History. It achieves only 13% on DexArt (vs. PF-DAG's 73%), confirming that simple history averaging is insufficient for multi-modal tasks.
  * **Visual Justification:** We provide side-by-side videos showing that VQ-VAE enables dynamic reactivity, whereas K-Means results in rigid behaviors. This qualitative difference justifies the design choice despite similar quantitative success rates in simulation.

#### **Reviewer u2ZP (Score: 8) – Theoretical Bounds**

* **Primary Concern:** The reviewer points out that our MSE bound proof assumes a perfect classifier.
* **Our Resolution:**
  * **Theoretical Refinement:** We provide a rigorous decomposition (added to **Appendix**) showing the trade-off: Single-stage policies suffer from guaranteed Inter-mode Variance ($V_{inter}$) (catastrophic mode collapse), whereas PF-DAG trades this for Classification Error ($E_{classify}$). We argue that minimizing the catastrophic error is preferable for task success.

---

### Meta-Review · Area_Chair_t6H8 · 2026-01-12

**Summary:**

The main concerns from reviewers centered on four key areas: 1) novelty and positioning relative to hierarchical approaches, 2) theoretical claims about the MSE bound, 3) empirical claims regarding marginal improvements from key components, and 4) architectural details and completeness of comparisons. Reviewers noted that the paper's core contribution of decoupling coarse action consistency from fine-grained variations was compelling but required stronger justification against existing hierarchical methods. The theoretical claim about achieving a "strictly lower MSE bound" was questioned as it assumed a perfect classifier, and reviewers questioned whether the improvements from VQ-VAE and MeanFlow were truly significant. The authors addressed these concerns by adding new experiments (QueST, RNR-DP, BC-H), providing a more rigorous theoretical analysis, clarifying the distinct contributions of PF-DAG, and adding quantitative stability metrics (jerk).

**Reviewer Concerns:**

Addressed:
- Reviewer u2ZP (original score 8): The MSE bound proof was clarified to show a trade-off between inter-mode variance and classification error, rather than a strict lower bound. The authors provided a rigorous decomposition in the appendix.
- Reviewer 6d45 (original score 4): The marginal improvements concern was addressed with ablation studies showing removing Primary Mode policy drops performance from 0.72 to 0.56 and removing MeanFlow drops it to 0.01. The comparison to RNR-DP was added, showing PF-DAG outperforms it. Stability metrics (jerk) were added to quantify trajectory smoothness.
- Reviewer YsMa (original score 4): The lack of comparison against hierarchical approaches was addressed by adding QueST comparison (PF-DAG 70% vs. QueST 26% on MetaWorld) and adding a new "Hierarchical and Residual Policies" subsection (Section 2.4). The architectural differences and fairness of comparisons were clarified.
- Reviewer 4ven (original score 6): The comparison against history-based baselines was addressed by adding BC-H (PF-DAG 0.73 vs. BC-H 0.13 on DexArt). The real-world evaluation details were clarified with additional hardware descriptions and failure analyses.

Still Outstanding:
- The concern about whether the two-stage approach is necessary rather than just a variant of hierarchical methods was addressed through strong evidence (ablation studies and new baselines), but the fundamental question about whether simpler methods could achieve similar results remains somewhat open. However, the ablation studies provide good evidence that the two-stage design is critical for the final performance.

**Reviewer Scores:**

- Reviewer u2ZP (Score: 8): Would likely remain at 8. The theoretical clarification was substantial, but the reviewer already gave a strong positive review.
- Reviewer 4ven (Score: 6): Would  be at least 6. The addition of BC-H baseline and better discussion of related work strengthens the paper.
- Reviewer 6d45 (Score: 4): Would likely increase to 6. The ablation studies, new comparison to RNR-DP, and stability metrics address the main concerns.
- Reviewer YsMa (Score: 4): Would likely increase to 6. The new comparisons, added citations, and clearer distinction from other hierarchical approaches address the main concerns about novelty.

The authors have substantially addressed the reviewers' concerns with new experiments, theoretical clarifications, and better contextualization of their work. Given the improvements made in the rebuttal, the paper meets the threshold for acceptance.

---

### Decision · Program_Chairs · 2026-01-26

Accept (Poster)